# A new method for the quantification of ambient particulate-matter emission fluxes

**Stergios Vratolis[1], Evangelia Diapouli[1], Manousos I. Manousakas[2], Susana Marta Almeida[3], Ivan Beslic[4], Zsofia Kertesz[5], Lucyna Samek[6], and Konstantinos Eleftheriadis[1]**

[1]ENvironmental Radioactivity & Aerosol Technology for atmospheric & Climate ImpacT Lab, INRASTES, NCSR Demokritos, 15310 Ag. Paraskevi, Attica, Greece
[2]Laboratory of Atmospheric Chemistry, Paul Scherrer Institute, Villigen-PSI, 5232, Switzerland
[3]Department of Nuclear Sciences and Engineering & C2TN, Instituto Superior Técnico, Universidade de Lisboa, Bobadela, Portugal
[4]Environmental Hygiene Unit, Institute for Medical Research and Occupational Health, Zagreb, 10000, Croatia
[5]Laboratory for Heritage Science, Institute for Nuclear Research (ATOMKI), Bem tér 18/C, Debrecen, 4026, Hungary
[6]Faculty of Physics and Applied Computer Science, AGH University of Science and Technology, ul. Mickiewicza 30, 30-059, Kraków, Poland

**Correspondence:** Stergios Vratolis (vratolis@ipta.demokritos.gr)

**Abstract.** An inversion method has been developed in order to quantify the emission fluxes of certain aerosol pollution sources across a wide region in the Northern Hemisphere, mainly in Europe and western Asia. The data employed are the aerosol contribution factors deducted by positive matrix factorization (PMF) on a $PM_{2.5}$ chemical composition dataset from 16 European and Asian cities for the period 2014 to 2016. The spatial resolution of the method corresponds to the geographic grid cell size of the Lagrangian particle dispersion model (Flexible Particle Dispersion Model, FLEXPART, $1° \times 1°$) which was utilized for the air mass backward simulations. The area covered is also related to the location of the 16 cities under study. Species with an aerodynamic geometric mean diameter of 400 nm and 3.1 μm and a geometric standard deviation of 1.6 and 2.25, respectively, were used to model the secondary sulfate and dust aerosol transport. Potential source contribution function (PSCF) analysis and generalized Tikhonov regularization were applied so as to acquire potential source areas and quantify their emission fluxes. A significant source area for secondary sulfate on the east of the Caspian Sea is indicated, when data from all stations are used. The maximum emission flux in that area is as high as $10 \times 10^{-12}\,\mathrm{kg\,m^{-2}\,s^{-1}}$. When Vilnius, Dushanbe, and Kurchatov data were excluded, the areas with the highest emission fluxes were the western and central Balkans and southern Poland. The results display many similarities to the $SO_2$ emission maps provided by the OMI-HTAP (Ozone Monitoring Instrument-Hemispheric Transport Air Pollution) and ECLIPSE (Evaluating the Climate and Air Quality Impacts of Short-Lived Pollutants) databases. For dust aerosol, measurements from Athens, Belgrade, Debrecen, Lisbon, Tirana, and Zagreb are utilized. The west Sahara region is indicated as the most important source area, and its contribution is quantified, with a maximum of $17.6 \times 10^{-12}\,\mathrm{kg\,m^{-2}\,s^{-1}}$. When we apply the emission fluxes from every geographic grid cell ($1° \times 1°$) for secondary sulfate aerosol deducted with the new method to air masses originating from Vilnius, a useful approximation to the measured values is achieved.

## 1   Introduction

Atmospheric aerosol particles affect air quality, human health, atmospheric visibility, and the climate (Laden et al., 2006; Lohmann and Feichter, 2005; Pope and Dockery, 2006; Ghosh et al., 2021; Burkart et al., 2022; Pandey et al., 2021; WHO, 2021). In order to identify and quantify aerosol sources and corresponding source areas, significant effort is required by the Scientific Community. When this information is acquired, measures can be applied so as to improve air quality. Source apportionment methods are widely used for air quality management, as they provide information on the relationship between air pollutant sources and their concentrations. The quantification of air pollution sources, both in terms of their sectorial and spatial origins, constitutes an essential step in the air quality management process (Wesseling et al., 2019).

In order to find the source areas for the pollution sources as identified by Almeida et al. (2020) we followed the potential source contribution function analysis (PSCF) (Eleftheriadis et al., 2009) and a discrete, deterministic approach (Tikhonov regularization; Tikhonov et al., 1995). Discrete, deterministic approaches have a long and distinguished history in geophysics. The major advantage of these methods is their computational efficiency, with costs governed by the number of discrete basis functions used. This limits the scale of the inference task to suit available resources but imposes strong assumptions about the properties of the model sought: we assume that it can be well-represented using the chosen set of basis functions. A drawback of any deterministic approach is the presumption that there is a single "answer" that can explain observations. In many cases, this cannot be true: available data plainly lack the sensitivity required to properly constrain all components within the basis function expansion. This motivates strategies that seek to identify the full range of models that might be compatible with observations (Park et al., 2018).

This study aims to introduce a two-step method for the quantitative estimation of emissions from geographic areas using in situ stations' measurement data. In the first step, the PSCF analysis for each measurement station is produced for the target species. Based on the results, we evaluate whether at a measurement station the target species are mainly transported or locally produced. In the second step, including only stations for which the target species are transported, we employ the Tikhonov regularization method in order to acquire emission fluxes from each geographic source area. The use of this method can reduce the uncertainty in emission fluxes, especially from those areas in which the emission inventories have high uncertainty. Numerous source apportionment studies have been conducted at many European and Asian cities in the past, and this method can identify the source areas of transported aerosols and quantify their emissions.

In the present work no a priori information was used, and a smooth solution was sought. The smooth solution is justi-fied by the fact that $SO_2$ emissions are gradually converted to secondary sulfate aerosol as they travel along with the air masses (Seinfeld and Pandis, 1998). This process takes many hours, covering probably more than one geographic grid cell ($1° \times 1°$). Dust aerosol possibly originates from multiple neighboring cells (i.e., in North Africa) and therefore a smooth solution is suitable for this case too.

It is important to note that the emission fluxes retrieved are subject to air mass transport paths, atmospheric conditions, and atmospheric chemistry. In other words, if a geographic grid cell emits a pollutant but air mass transport does not allow these emissions to reach any of the measurement stations in the study, this cell will not be attributed the emission flux that it has. For species like secondary sulfate, identical precursor gases emission fluxes could lead to different aerosol concentrations, depending on atmospheric conditions and chemistry. It is also possible that locally produced aerosol (that is within the station grid cell) cannot be correctly associated with residence time in the grid cell. That is because emission fluxes in the vicinity of the measurement stations have a very small residence time until they arrive at the station and have a very high impact on the measured concentration. Despite these potential problems, the information on specific geographic grid cells that actually impact the measurement stations area is focused on where mitigation measures for long-range transport must be applied.

From now on, we refer to "source apportioned concentration by positive matrix factorization (PMF)" as "concentration" and to "geographic grid cell source area emission fluxes" as "emission fluxes". NE corresponds to the northeast, NW to the northwest, SE to the southeast, and SW to the southwest.

## 2   Materials and methods

### 2.1   Particulate-matter (PM) sampling stations and filter analysis

More than 2200 $PM_{2.5}$ samples were collected in urban and suburban background stations from 16 European and Central Asian cities as presented in Fig. 1 (Tirana, Zagreb, Chisinau, Athens, Skopje, Debrecen, Banja Luka, Sofia, Belgrade, Kraków, Nikšić, Kurchatov, Dushanbe, Vilnius, Lisbon, Ankara). The Ankara and Belgrade stations are reported as suburban background by Almeida et al. (2020), while all other stations are reported as urban background. Sampling was performed mostly in 24 h periods, every third day, between January 2014 and December 2016. Particles were sampled on PTFE, polycarbonate, cellulose nitrate, cellulose, and quartz filters by means of low- and medium-volume samplers.

Before and after sampling, filters were weighed in the laboratories located in each city by means of a microbalance using the procedure described in EN12341 (1998). Filters were subsequently analyzed by several analytical techniques for

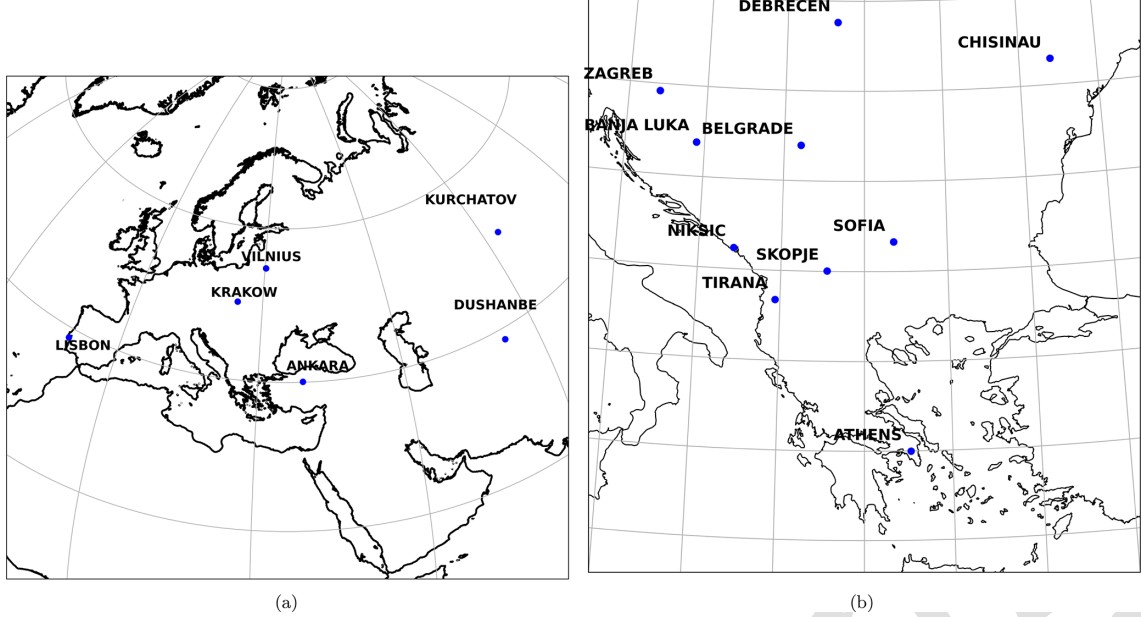

**Figure 1.** Urban background and suburban background measurement stations included in the study. The Ankara, Dushanbe, Vilnius, Kraków, Kurchatov, and Lisbon sampling locations are depicted in panel **(a)**, while the Athens, Banja Luka, Belgrade, Chisinau, Debrecen, Nikšić, Skopje, Sofia, Tirana, and Zagreb sampling locations are depicted in panel **(b)**.

the determination of major and trace elements, elemental and organic carbon, black carbon, and water-soluble ions.

The positive matrix factorization receptor model (EPA PMF 5.0; Brown et al., 2015) was applied and sources were acquired for each city.

Due to the high number of cities involved in this work, it was not possible to fully harmonize the methods used, which introduces a level of uncertainty in the results obtained and especially in their comparison. Source apportionment (SA) of $PM_{2.5}$ was performed by receptor modeling that is based on the mass conservation principle. Further uncertainties to the source apportionment results were introduced by the fact that at the stations of Chisinau, Sofia, Nikšić, Lisbon, Ankara, and Vilnius only 50 filter samples are available. We have not applied PMF to less than 50 samples in any of the cities. Fifty samples have been recorded as the minimum necessary for a meaningful source apportionment analysis according to Manousakas et al. (2017b) and Johnson et al. (2011). Having said that, it has been identified in the past that small datasets (with the number of samples close to 50) pose an extra challenge when used for PMF because the solution is strongly affected by rotational ambiguity and the overall uncertainty is increased. Before using the results, we have fully assessed the uncertainty in the SA solution using the enhanced tool offered by EPA PMF 5.0. The uncertainty was within acceptable limits. We included these measurements because they are valuable, as aerosol data from these areas are scarce and, also, because including them would diversify

the origin of air masses used in the identification of source areas and emission fluxes, making our results more precise.

More details can be obtained in Almeida et al. (2020), where the measurement stations, $PM_{2.5}$ analysis techniques used, and PMF results are described in detail.

## 2.2 Flexible Particle Dispersion Model (FLEXPART)

The Flexible Particle Dispersion Model (FLEXPART) was used in order to acquire residence times over geographic grid cells (Stohl et al., 2005, 2009). These residence times indicate how sensitive the measurements at a station are to emissions occurring at each geographic grid cell. FLEXPART runs account for grid-scale wind as well as for turbulent and mesoscale wind fluctuations. Drift correction, to prevent accumulation of the released computational particles, and density correction, to account for the decrease in air density with height, were both applied. Twenty-day backward runs with the release of $4 \times 10^4$ air parcels every 3 h beginning from each station were produced. The residence time for each of these air parcels over each grid cell was calculated. Then the average was taken for all air parcels for each grid cell. This is the sensitivity for each 3 h. We then sum these 3 h sensitivities so as to correspond exactly to each filter sampling time. The aerosol species carried by the air parcels were secondary sulfate (400 nm aerodynamic geometric mean diameter, 1.6 standard deviation) and dust (3.1 µm aerodynamic geometric mean diameter, 2.25 standard deviation). Wet and dry deposition of these species was also included in the model. Residence times in each grid cell, for a height range from 0 to

500 m above ground level (a.g.l.), are used for this study. The height was chosen so as to include sources within the boundary layer for all geographic grid cells. Chemical reactions were not simulated in the backward runs.

The properties of the species were chosen based on the work published by Gini et al. (2022), where an 11-stage low-pressure Berner impactor was used. The Berner impactor aerodynamic diameter sizes range from 0.03 to 13.35 µm at a flow rate of 26 L min$^{-1}$. Gini et al. (2022) determined the elemental composition of the collected samples by energy dispersive X-ray fluorescence spectroscopy (XRF).

In the case of secondary sulfate, it is important to keep in mind that $SO_2$ is the primary emitted species and secondary sulfate is produced in the atmosphere through chemical reactions in the gas and liquid phase. In order to calculate the uncertainty that this error induces in the calculated footprint, we refer to residence times in the atmosphere reported by Seinfeld and Pandis (1998, p. 66). The $SO_2$ mean residence time reported due to dry deposition is 60 h, its residence time due to wet deposition is 100 h, and its residence time due to transformation to secondary sulfate is 80 h. The resulting $SO_2$ residence time due to wet and dry deposition is 37.5 h, while if we also include the transformation to sulfate the overall mean residence time is 25 h. The corresponding wet- and dry-deposition residence time indicated for secondary sulfate is 80 h. Therefore, in such a case, $SO_2$ deposits (wet and dry deposition) twice as fast as secondary sulfate. These calculations correspond to the midlatitudes (45–65° N) according to Rodhe (1978). FLEXPART is provided with a secondary sulfate aerosol particle size distribution, and it compensates for wet and dry deposition as it follows the species backward in time. The error in the calculation of the residence time in each geographic grid cell is mainly due to not accounting for the enhanced deposition of $SO_2$ for 1–2 d just after emission. But this enhanced wet and dry deposition for $SO_2$ should be applied only for a small fraction of the travel time. The mean error in residence time due to this discrepancy is expected to be close to 10 %.

It should be noted that we do not present emission fluxes of $SO_2$ but the origin of secondary sulfate aerosol measured at each station if it was produced as such in the emitting grid cell. Therefore we report the combined effect of $SO_2$ emissions, air mass transport, and environmental conditions that produce the secondary sulfate aerosol measured in the stations participating in the study. That is why we believe that the fluxes derived cannot be applied to very distant measurement stations, whose environmental conditions might be very different from the stations in the study. Also, the estimated error was calculated based on values derived for the midlatitudes.

Since secondary sulfate has a mean residence time of 80 h in the atmosphere, as reported by (Seinfeld and Pandis, 1998), we expect that most of the secondary sulfate aerosol measured at each station has been produced in the atmosphere within the previous week. This would probably corre-spond to regional transport, not global. We expect that most of the dust aerosol measured at each station would be regional since it has a much larger aerodynamic mean diameter of 3.1 µm, leading to a much faster deposition velocity. In any case, both species are followed backward in time for 20 d and residence times are attributed for all geographic grid cells. However, for the inversion we use the residence times in each cell for the area between a latitude of −30 to 90° N and a longitude from −40 to 140° E.

## 2.3  Tikhonov regularization

We are concerned with the solution of minimization problems of the form

$$\min \|\mathbf{A}x - b\| \quad x \in R^n, \tag{1}$$

where $\|\cdot\|$ denotes the Euclidean norm, $\mathbf{A} \in R^{m \times n}$ is an ill-conditioned matrix, and the data vector $b \in R^m$ is contaminated by an unknown error $e \in R^m$ that may stem from measurement inaccuracies and discretization error (Park et al., 2018). Thus, $b = b_{\text{exact}} + e$. We are interested in computing the solution $x_{\text{exact}}$ of a minimal Euclidean norm of the least-squares problem with the error-free data vector

$$\min \|\mathbf{A}x - b_{\text{exact}}\| \quad x \in R^n, \tag{2}$$

associated with Eq. (2). The desired solution $x_{\text{exact}}$ will be referred to as the exact solution. Since $b_{\text{exact}}$ is not known, we seek to determine an approximation of $x_{\text{exact}}$ by computing a suitable approximate solution of Eq. (2).

Due to the ill conditioning of matrix $\mathbf{A}$ and the error $e$ in the data vector $b$, a straightforward solution of the least squares problem (2) generally does not give a meaningful approximation of $x_{\text{exact}}$. Therefore, the minimization problem of Eq. (2) is commonly replaced by a penalized least-squares problem of the form

$$\min \left\{ \|\mathbf{A}x - b\|^2 + \lambda^2 \|\mathbf{L}(x - x_0)\|^2 \right\} \quad x \in R^n. \tag{3}$$

This replacement is known as Tikhonov regularization. The parameter $\lambda \geq 0$ is the regularization parameter that balances the influence of the first term (the fidelity term) and the second term (the regularization term), which is determined by the regularization matrix $\mathbf{L} \in R^{p \times n}$. Here $p$ is an arbitrary positive integer. $x_0$ represents our a priori knowledge on the solution.

The purpose of the regularization term is to damp undesired components of the minimal-norm least-squares solution of Eq. (1). The minimization problem (3) is said to be in standard form when $\mathbf{L}$ is the identity matrix $\mathbf{I}$; otherwise the minimization problem is said to be in general form. We are interested in Tikhonov regularization in general form because for a suitable choice of regularization matrix $\mathbf{L} \neq \mathbf{I}$, the solution of Eq. (3) can be a much better approximation of $x_{\text{exact}}$ than the solution of Eq. (3) with $\mathbf{L} = \mathbf{I}$. A smooth solution is obtained when the $\mathbf{L}$ matrix requires that the difference between

two neighboring cells is minimal. In other words, when the regularization matrix $\mathbf{L}$ is the first-order discrete derivative operator, it imposes smoothness on the solution (Donatelli and Reichel, 2014). In our particular case, each row of the $\mathbf{A}$ matrix corresponds to FLEXPART sensitivity (residence time in each grid cell) for each filter measurement, and each column of the $\mathbf{A}$ matrix corresponds to a specific geographic grid cell sensitivity for all filter measurements. $\boldsymbol{b}$ corresponds to the actual species mass concentration for each filter, while $\boldsymbol{x}$ is the emission flux from each geographic grid cell. In other words, we try to extract information associated with residence time in each grid cell for each filter measurement.

We expect that uncertainties associated with the PM$_{2.5}$ measurements, chemical analysis, and PMF model application will also be attributed as unknown error $\boldsymbol{e}$ in the regularization term. Cavalli et al. (2016) report a positive sampling artifact of 0.4 to $2.8\,\mu g\,C\,m^{-3}$ for PM collection on quartz fiber filters corresponding to 14 %–70 % of the total carbon collected. Viana et al. (2006) report that approximately 14 % of the PM$_{2.5}$ mass may result from the adsorption of gaseous organic and inorganic compounds onto the filter or the particles already collected on it (positive artifact). They also state that prolonged sampling times may lead to greater negative artifacts (i.e., the loss of semi-volatile organic compounds and of ammonium nitrate). The uncertainty in the XRF, elemental carbon (EC), organic carbon (OC), and ion chromatography (IC) measurements range between less than 10 % (IC) and up to 20 % (XRF) (Manousakas et al., 2017a; Panteliadis et al., 2015; Mantas et al., 2014; Vratolis et al., 2018). According to the AIRUSE 2016 EU project final report (Deliverable B2.4; IDAEA, 2016; Amato et al., 2016; Diapouli et al., 2017), the PMF results' standard error was estimated for the secondary sulfate source to be below 10 %, while the dust source standard error ranged from below 5 % to 40 % (PM$_{2.5}$ filters). An overall uncertainty approximating 30 % in the results obtained from the filter analysis and species concentration for each city is therefore expected.

When no a priori information is available, the assumption in the Tikhonov regularization equation is that $\boldsymbol{x}_0$ is a vector of zeros. We seek in our case a smooth solution, requesting that emission fluxes of neighboring cells have differences close to 0, while at the same time the measured concentrations are reconstructed by the solution. Solutions with small emission flux absolute values have smaller differences in neighboring cells than solutions with large emission flux absolute values. This imposes solutions with emission fluxes as small as possible, leading to the underestimation of measured values. The underestimation is relevant to how important the regularization term in Eq. (3) is. A perfect fit between the measured and modeled data is achieved when the regularization parameter $\lambda$ is equal to 0. As we mentioned in the previous paragraph, an overall uncertainty approximating 30 % is expected. In order to regularize such an uncertainty level a large regularization parameter $\lambda$ is required, thus leading to a significant underestimation of the model results. We have to keep in mind that if $\lambda$ is close to 0, we perfectly reconstruct the measured concentrations, which include a large error due to the aforementioned reasons. As the inverse process is not linear, such an approach would result in very large errors in the estimation of emission fluxes in each grid cell.

A secondary sulfate aerosol species was identified in 14 out of 16 cities in the study, and therefore the two cities without this species (Ankara, Lisbon) were excluded from the analysis. In a small number of samples in the 14 cities included in the study, negative concentrations were identified. These samples were excluded from the dataset used in the Tikhonov regularization. Dust aerosol concentration was identified in 16 cities. Nevertheless, after the PSCF analysis for dust aerosol, only six cities were included in the Tikhonov regularization dataset. That is because the PSCF analysis indicated that most of the dust aerosol identified was of local origin (dust resuspension). Filter samples that had negative dust concentrations were also excluded.

## 2.4 $L$-curve method

Commonly, if only a single regularization parameter needs to be determined, the norms of model and residuals are plotted against one another so as to give an $L$ curve. This name comes from the curve's characteristic shape, and the preferred regularization parameter is then chosen by identifying the "elbow" of the curve. The strategy is justified by the principle of Occam's razor, which advocates reliance on the simplest (in the present context, smallest) model that can explain observations (Valentine and Sambridge, 2018; Hansen, 1992).

## 2.5 Potential source contribution function (PSCF)

Twenty-day backward FLEXPART runs were used to acquire the residence time over each geographic cell for each measurement and for all stations. For each cell the PSCF ratio was calculated.

$$\text{PSCF}_{i,j} = \text{weight}_{i,j} \times m_{i,j}/n_{i,j}, \tag{4}$$

where $m_{i,j}$ is the sum of residence times (sensitivity) in a cell for concentrations higher than the 90th percentile and $n_{i,j}$ is the sum of residence times for all measurements. The indexes $i$ and $j$ correspond to latitude and longitude of each grid cell. PSCF$_{i,j}$ is the measure of probability of a grid cell ($1° \times 1°$) to contribute to the concentration of the pollutant measured at the receptor site considered (Perrone et al., 2018). In order to acquire the weight factor used for each cell, total residence times in cells were divided in percentiles. The weight coefficients 0.25, 0.5, and 0.75 were used for cells with total residence times up to the 25th, 50th, and 75th percentiles, respectively.

We apply the PSCF analysis for each measurement station and each aerosol species. The information that we use is the overall residence time for all filters in each station ($n_{i,j}$) and

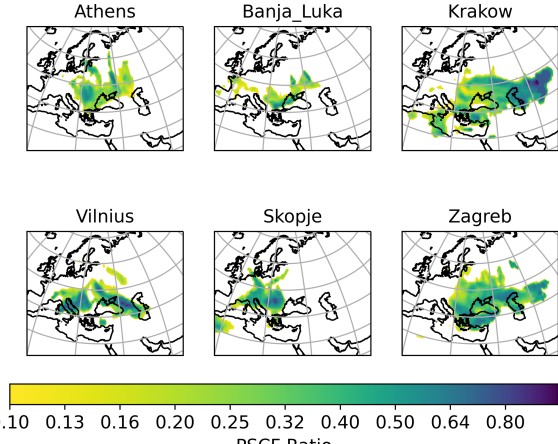

**Figure 2.** PSCF analysis for concentrations higher than the 90th percentile of secondary sulfate aerosol in Zagreb, Athens, Vilnius, Kraków, Banja Luka, and Skopje. The horizontal bar represents the PSCF ratio ($\text{PSCF}_{i,j} = \text{weight}_{i,j} \times m_{i,j}/n_{i,j}$).

the overall residence times in each grid cell for the filter measurements with the highest secondary sulfate or dust aerosol concentrations ($m_{i,j}$). In other words, we extract information from the sum of residence times for all filters and the sum of residence times for filters with the highest concentration (90th percentile). Grid cells with a very small residence time may result in PSCF with high uncertainty in the apparent high value. For large values of $n_{i,j}$, there is more statistical stability in the calculated value. Thus, to reduce the effect of small values of $n_{i,j}$, an empirically determined weight matrix is multiplied with the PSCF ratio to better reflect the uncertainty in the values for these cells (Polissar et al., 2001).

Twenty-day backward runs were used so as to assess species with high residence times in the atmosphere, like Sahara dust.

## 3 Results and discussion

### 3.1 Secondary sulfate aerosol

The secondary sulfate concentration identified in each station was simulated by an aerosol log-normal distribution with an aerodynamic geometric mean diameter of 400 nm and a standard deviation of 1.6. From each station the aerosol mass was released every 3 h (within $4 \times 10^4$ discrete finite air masses) and followed backward in time for 20 d. The result obtained by FLEXPART was the residence time in each geographic cell. In Fig. 2, the PSCF results for Zagreb, Athens, Kraków, Skopje, Vilnius, and Banja Luka are displayed. These cities were chosen as the areas indicated by their PSCF correspond to high emission fluxes according to emission maps (OMI-HTAP, ECLIPSE). In Appendix A1 we display the PSCF results for the rest of the cities for which a secondary sulfate concentration was identified.

In Fig. 2, Athens indicates the center of the Balkans and eastern Europe as source areas. Kraków points mainly to the area east of the Caspian Sea. Banja Luka's secondary sulfate main origin is the Volga region and the eastern Balkans. Vilnius's secondary sulfate comes from Ukraine and the Balkans. Skopje's secondary sulfate stems from the north of the Balkans and northern Italy. A source area is also indicated in NW Africa. Zagreb indicates the central and eastern Balkans, the area around the Caspian Sea, and Asia Minor as source areas.

In two cities (Ankara, Lisbon), no secondary sulfate concentration was indicated by the PMF analysis. Therefore 14 out of the 16 cities could be included, namely Tirana, Zagreb, Chisinau, Athens, Skopje, Debrecen, Banja Luka, Sofia, Belgrade, Kraków, Nikšić, Kurchatov, Dushanbe, and Vilnius (around 2050 measurements). Our first approach was to apply the Tikhonov regularization to data from the six cities indicated by the PSCF analysis in Fig. 2. Then we applied regularization to all 14 cities.

We applied the Tikhonov regularization to all 14 cities with an identified secondary sulfate concentration, as we consider that secondary sulfate and its precursor gases are emitted from many source areas in both Europe and Asia, and we needed as many stations and measurements as possible in order to identify them.

In Fig. 3, the emission fluxes for secondary sulfate aerosol calculated by the Tikhonov regularization method for $1° \times 1°$ cells are presented. We used this resolution in the range of latitudes from $-30$ to $90°$ N and longitudes from $-40$ to $140°$ E. This corresponds to a $120 \times 160$ (19 200 unknown factors) emission cell matrix, a number much higher than the total number of measurements. It is important to keep in mind that not all species are measured at all stations, and even when a species exists at a station, it may not be present in all samples.

The result for the 6 cities (1069 measurements) is depicted in Fig. 3a, while the result for 14 cities is displayed in Fig. 3b.

The $SO_2$ emission inventory of OMI-HTAP is also displayed in Fig. 3c (Liu et al., 2018). It includes the non-energy emissions (from industry, residential, and transportation) and the energy emissions. Note that aviation and shipping emissions are not included in the OMI-HTAP inventory. The high-emission grid cells of the Tikhonov regularization solution for 14 cities are indicated by shaded areas. We observe that there are high emission fluxes in the $SO_2$ OMI-HTAP map in the indicated areas.

We also observe in Fig. 3 that the areas indicated by the Tikhonov regularization solution for 14 cities (Fig. 3b), namely the central and western Balkans, southern Poland, and the area east of the Caspian Sea, are apparent also in the ECLIPSE $SO_2$ database map (Klimont et al., 2017). Again, we indicate these areas by adding a shaded oval. The ECLIPSE $SO_2$ database includes energy production, industry, oil and gas flaring, transport, shipping, agriculture, and residential and waste emissions. As already mentioned ear-

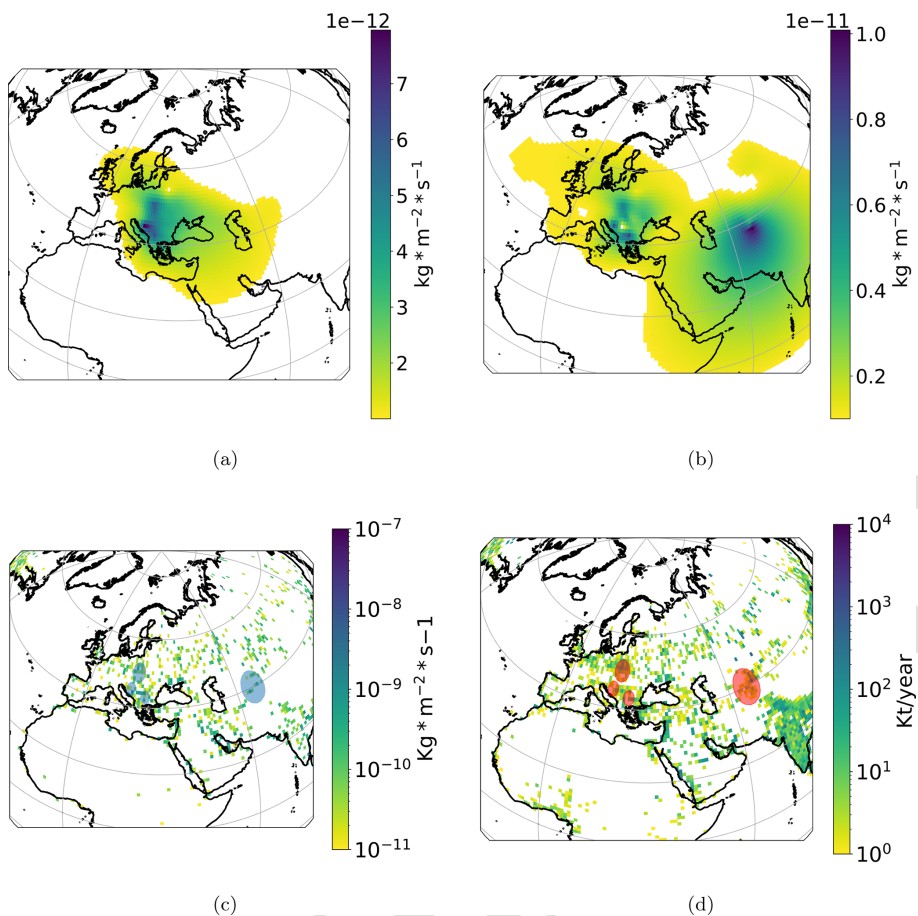

**Figure 3.** Secondary sulfate aerosol Tikhonov regularization, $1° \times 1°$ emission fluxes, and OMI-HTAP and ECLIPSE emission maps. In panel **(a)** the solution for the six cities indicated by PSCF analysis (Zagreb, Athens, Vilnius, Kraków, Banja Luka, Skopje) is demonstrated. Panel **(b)** depicts the Tikhonov regularization emission flux solution when data from 14 stations are included. In panel **(c)** the OMI-HTAP emissions map during 2015 for $SO_2$ is demonstrated with units of $kg\,m^{-2}\,s^{-1}$. In panel **(d)** the ECLIPSE V6b database for $SO_2$ emissions during 2015 is presented with units of $kt\,yr^{-1}$. Grid cells attributed high emission fluxes by the Tikhonov regularization are indicated in the emission inventories maps by shaded areas.

lier, a high level of uncertainty is expected in our input data. We excluded negative secondary sulfate concentration measurements from the calculations, due to their high uncertainty. In Fig. 3b, which corresponds to the solution for all available data (14 stations), the highest values are as follows.

For the area east of the Caspian Sea, the maximum value is $10 \times 10^{-12}\,kg\,m^{-2}\,s^{-1}$ for latitude 37–38° N and longitude 67–68° E. In the OMI-HTAP $SO_2$ map, the maximum value in the area is at 39–40° N and 65–66° E, with a value of $7.7 \times 10^{-10}\,kg\,m^{-2}\,s^{-1}$.

For the area in the western Balkans, the maximum value is $7.8 \times 10^{-12}\,kg\,m^{-2}\,s^{-1}$ for latitude 44–45° N and longitude 16–17° E. In the OMI-HTAP $SO_2$ map, the maximum value in the area is at 44–45° N and 18–19° E, with a value of $9.2 \times 10^{-10}\,kg\,m^{-2}\,s^{-1}$.

For the area in Poland, the maximum value is $6.1 \times 10^{-12}\,kg\,m^{-2}\,s^{-1}$ for latitude 49–50° N and longitude 19–20° E. In the OMI-HTAP $SO_2$ map, the maximum value in the area is at 51–52° N and 19–20° E, with a value of $5.3 \times 10^{-10}\,kg\,m^{-2}\,s^{-1}$.

For the area in the central Balkans, the maximum value is $8.3 \times 10^{-12}\,kg\,m^{-2}\,s^{-1}$ for latitude 42–43° N and longitude 20–21° E. In the OMI-HTAP $SO_2$ map, the maximum value in the area is at 44–45° N and 18–19° E, with a value of $9.3 \times 10^{-10}\,kg\,m^{-2}\,s^{-1}$.

In the solution for 6 cities (Fig. 3a), very similar values to the ones for 14 cities were acquired in the areas of the western and central Balkans and Poland. We expect that the hotspot areas in the Tikhonov regularization solution are the most important for the transported secondary sulfate for the cities in the study, even though the calculated emission flux values might differ from the ones in emission inventories.

We also produced two more emission flux results: one including only measurements from Zagreb (around 600 measurements) and one including all participating European cities except Vilnius (around 1800 measurements). The first

result is indicative of using a dataset from just one measurement station. In the second result we exclude Vilnius, Dushanbe, and Kurchatov data. Dushanbe and Kurchatov are situated at a significant distance from other stations, outside the region of Europe. Vilnius on the other hand is on the edge of the area that is covered by European stations. This result was produced as we wish to evaluate whether we could predict secondary sulfate concentration in Vilnius.

The result for Zagreb (Fig. 4b), due to the small number of measurements (562 samples with positive secondary sulfate source values) used in relation to 19 200 unknown factors, lacks specificity, indicating Poland and eastern Europe in general as the main source area. The central and western Balkans also have a high impact on Zagreb. We included the emission flux results for Zagreb as it was the station with the largest number of filter samples in the study. This case represents the results we could expect when we use data from a single station.

When we compare the OMI-HTAP emission map for $SO_2$ (Liu et al., 2018) to the emission map acquired by the Tikhonov regularization for the investigated European cities excluding Vilnius (Fig. 4a), we observe many similarities. The hotspots in the Balkans and southern Poland are apparent in both maps.

We also find similarities between the PSCF analysis (Fig. 2) and the regularization result when 14 cities are included. In particular, the area east of the Caspian Sea appears to contribute significantly in the PSCF performed for Zagreb and Kraków as well as in the Tikhonov regularization solution.

The PSCF result for Dushanbe (Fig. A1) indicates the area east and SE of the Caspian Sea as potential sources. This is also apparent in the solution when we include all stations and the OMI-HTAP emission map for $SO_2$. Central Asia, comprising Kazakhstan, Kyrgyzstan, Tajikistan, Uzbekistan, and Turkmenistan, has developed rapidly in terms of population, industrialization, and urbanization over the past few decades, accompanied by increased anthropogenic emissions. These emissions, along with regional and local dust, are often subject to long-range atmospheric transport by westerlies toward the Tian Shan and the Tibetan Plateau. Biomass burning is a significant contributor to primary organic carbon emissions (Chen et al., 2022).

The center of the Balkans appears as a source area according to the PSCF for Zagreb and Athens as well as the Tikhonov regularization solution for 14 cities (Fig. 3b).

In the Tikhonov regularization result for the six cities (Fig. 3a), resulting emission fluxes for all grid cells were positive. In the other three Tikhonov regularization results (Figs. 3b, 4a, b), we allowed for small negative emission fluxes ($-5 \times 10^{-13}$ kg m$^{-2}$ s$^{-1}$).

When we compare the modeled concentrations for secondary sulfate using the solution for 14 cities to measured values at each station (Fig. 5b), the agreement is not good. This is probably due to uncertainties associated

with the data, the influence of the regularization term in Eq. (3), and the lack of a priori information, as explained in Sect. 2.3 (Tikhonov regularization). In Fig. 5b, the intercept is 3 µg m$^{-3}$ and the slope is close to 0.3.

The modeled concentration was acquired according to Eq. (5), following (Pisso et al., 2019).

$$\text{Model conc. (kg m}^{-3}) =$$

$$\sum_{\text{lat}=-30°}^{90°} \sum_{\text{long}=-40°}^{140°} \Big( \text{residence time}_{i,j} \text{ (s)}$$

$$\times \boldsymbol{x}_{\text{exact}-i,j}(\text{kg m}^{-2}\text{s}^{-1})/\text{height of 500 m} \Big) \quad (5)$$

In Fig. A4 (Appendix) we present the emission fluxes Tikhonov regularization solution (14 cities) for secondary sulfate aerosol during summer (April to September) and winter (October to March) months. In winter, as expected, emission fluxes have significantly higher values than summer. In summer the hotspot east of the Caspian Sea almost disappears, indicating that these emissions probably relate to heating. In southern Poland the hotspot is significantly reduced. Hotspots on western and central Balkans appear to have similar values in winter and summer, indicating that they possibly originate from power plants.

### The case of Vilnius

In Fig. 6, we compare the modeled and measured secondary sulfate aerosol concentration at Vilnius with the 1° resolution model. We used the result of the Tikhonov regularization when we excluded the data from Vilnius, Dushanbe, and Kurchatov to acquire the modeled values.

In general, the agreement between the modeled and the measurement data is relatively good. The agreement is not good for very low measured concentration values of secondary sulfate. The lowest PM$_{2.5}$ concentrations in the dataset are observed during August, September, until nearly the end of October. This could also be related to the beginning of the winter season, with increased emissions due to heating.

Vilnius station was chosen for the demonstration of the results as it is situated on the edge of the area that the rest of the European stations of the study cover.

### 3.2   Dust aerosol

The first step in order to identify potential source areas was to apply the PSCF analysis on all cities. Meaningful results, in the sense that the indicated potential source areas do indeed emit dust aerosol, were acquired only in the cases of Athens, Belgrade, Debrecen, Lisbon, Tirana, and Zagreb. In Fig. 7 the PSCF results at the 90th percentile for dust aerosol for these stations are displayed. Potential source areas for Athens were North Africa and the Middle East, for Belgrade NE Africa and the Middle East, for Debrecen western Africa

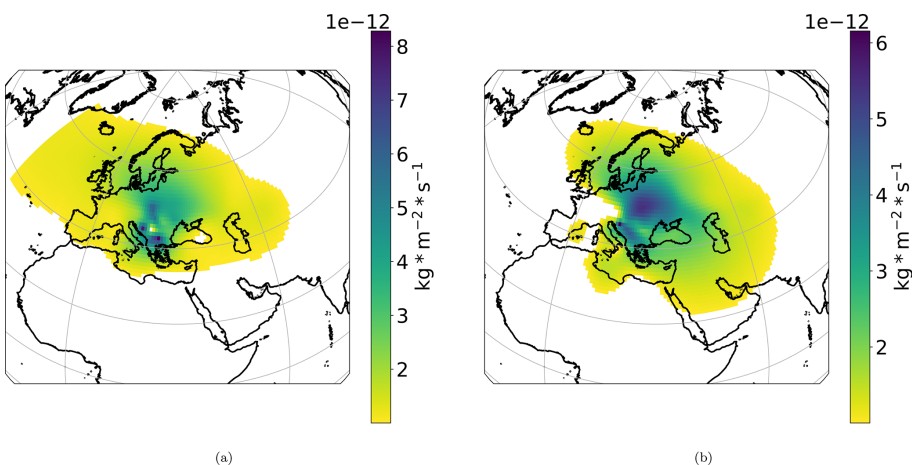

(a)                                                                (b)

**Figure 4.** Secondary sulfate aerosol Tikhonov regularization and $1° \times 1°$ emission fluxes for **(a)** European stations excluding Vilnius (Athens, Banja Luka, Belgrade, Chisinau, Debrecen, Nikšić, Skopje, Sofia, Tirana, Kraków, and Lisbon) and **(b)** Zagreb. The vertical bar corresponds to the emission fluxes of secondary sulfate aerosol from each geographic grid cell in $\mathrm{kg\,m^{-2}\,s^{-1}}$.

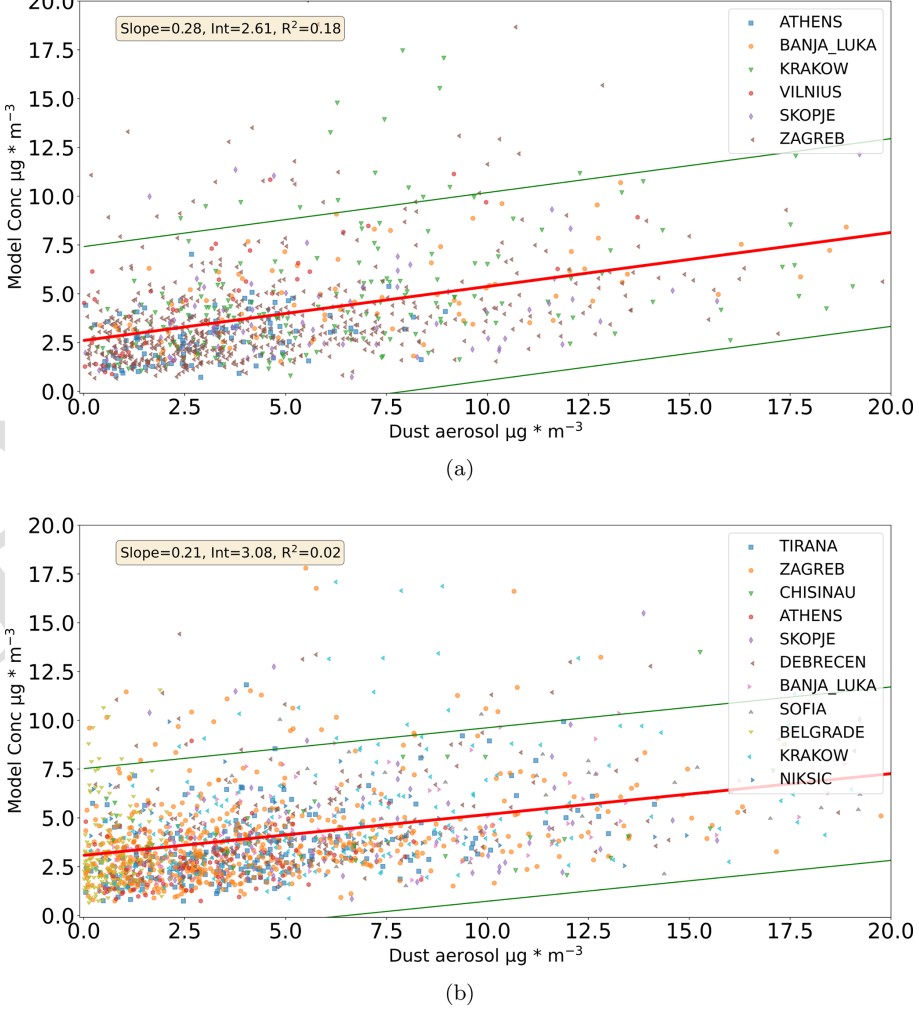

(a)

(b)

**Figure 5.** Comparison between the measured secondary sulfate and the modeled values based on the Tikhonov regularization solution for 6 cities in **(a)** and 14 cities in **(b)**. Regression line $\pm 2$ standard deviations is also depicted. The legend presents the measurement location.

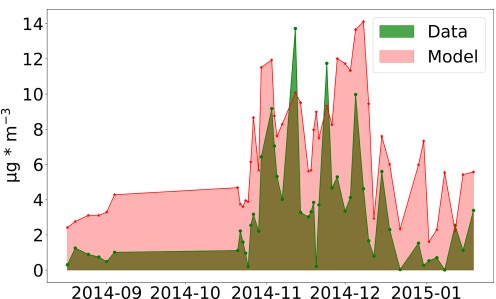

**Figure 6.** Comparison of modeled and measured secondary sulfate aerosol concentration at Vilnius. The emission flux solution used is the one acquired by Tikhonov regularization when Vilnius, Dushanbe, and Kurchatov data are excluded (data from Athens, Banja Luka, Belgrade, Chisinau, Debrecen, Nikšić, Skopje, Sofia, Tirana, Kraków, and Lisbon).

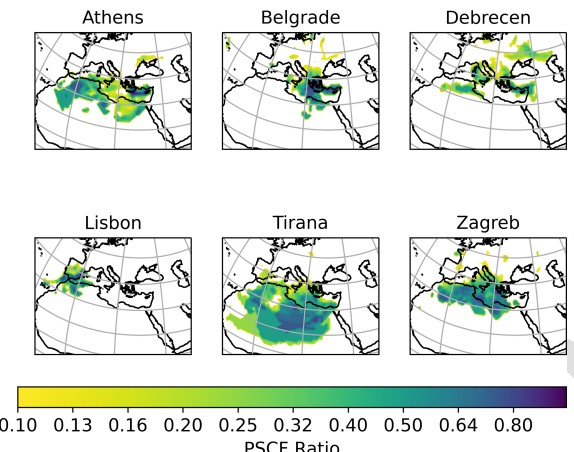

**Figure 7.** PSCF analysis for concentrations higher than the 90th percentile of dust aerosol in Athens, Belgrade, Debrecen, Lisbon, Tirana, and Zagreb. The horizontal bar represents the PSCF ratio ($\mathrm{PSCF}_{i,j} = \mathrm{weight}_{i,j} \times m_{i,j}/n_{i,j}$).

and the Middle East, for Lisbon western Africa, for Tirana North Africa, and for Zagreb North Africa. The PSCF results for the rest of the stations (Ankara, Dushanbe, Vilnius, Kraków, Kurchatov, Banja Luka, Chisinau, Nikšić, Skopje, Sofia) indicated that their dust aerosol was mainly of local origin (dust resuspension; please refer to Figs. A6 and A7 in the Appendix).

In the second step, in order to quantify the dust aerosol emitted from each geographic grid cell, the Tikhonov regularization was applied to the data from Athens, Belgrade, Debrecen, Lisbon, Tirana, and Zagreb, excluding negative values (1320 measurements were used).

In Fig. 7 the PSCF for the 90th percentile for dust aerosol is presented. In the PSCF subfigure for Tirana two paths can be seen: in the first path, winds from the Atlantic Ocean pass over NW Africa and then the Mediterranean Sea, subsequently reaching Tirana. In the second path, winds from the

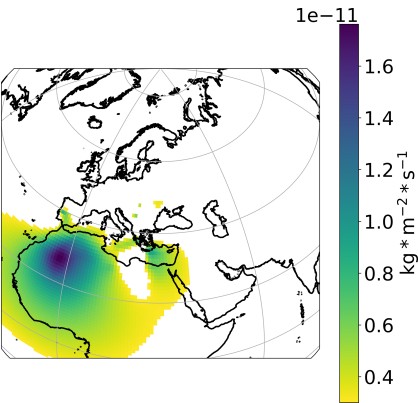

**Figure 8.** Dust aerosol Tikhonov regularization and $1° \times 1°$ emission fluxes for data from Athens, Belgrade, Debrecen, Lisbon, Tirana, and Zagreb. The vertical bar corresponds to the emission fluxes of dust aerosol from each geographic grid cell in $\mathrm{kg}\,\mathrm{m}^{-2}\,\mathrm{s}^{-1}$. The main source area depicted is NW Africa.

Atlantic Ocean pass over NW Africa and then NE Africa and the Mediterranean Sea, subsequently reaching Tirana. The second path is by far the prevailing one for the 90th percentile highest concentrations of dust aerosol for Tirana, as can be seen in Fig. A3 in the Appendix. It is important to keep in mind that the residence times depicted correspond to a height of up to 500 m so as to always be within the boundary layer. Therefore, while the dust load could be mainly picked up in both cases in NW Africa, due to longer residence times in NE Africa, this area could appear as the most probable to be the one that emits dust aerosol. This could be partly due to the fact that as the air masses travel over Africa at low altitude, wind speed is reducing due to higher friction over land in comparison to when they travel over the sea (Atlantic or Mediterranean). The air masses probably have higher speed over NW Africa, and this results in more dust being picked up in this area. Some dust aerosol could be picked up from NE Africa, and its origin could also be the Arabian Peninsula. This path is also evident in Fig. 8, where a weak emission area is indicated in NE Africa.

While for the PSCF analysis Tirana, Zagreb, and Belgrade indicate a high probability of NE Africa being a source area, this is not the case for the Tikhonov regularization result. In Fig. 8, the result indicates that NW Africa is by far the most significant dust aerosol source area for the six cities (Athens, Belgrade, Debrecen, Lisbon, Tirana, and Zagreb) whose data we used. NE Africa also has a hotspot in Fig. 8, but its contribution was significantly lower when the data from these six stations are combined. In the borders between Mauritania, Algeria, and Mali, the highest emission fluxes are identified (lat 27°, long −4°), which are as high as $17.6 \times 10^{-12}\,\mathrm{kg}\,\mathrm{m}^{-2}\,\mathrm{s}^{-1}$.

Stohl et al. (2009), referring to halocarbons, state that inaccuracies in model and data will in general cause their method to find solutions containing unrealistic negative

emissions that are larger than expected. In the linear framework, this cannot be prevented directly as positive definiteness is a nonlinear constraint. They also suggest an iteration method so that the sum of all negative emissions is less than 3‰ of the sum of the positive emissions. In our case with the dust aerosol we allow small negative emission values ($-2.5 \times 10^{-12}$ kg m$^{-2}$ s$^{-1}$) representing higher deposition velocities than calculated by the FLEXPART deposition scheme.

In Fig. 9 the comparison between the modeled data using the Tikhonov regularization solution for dust and measured concentrations is presented. In this case we have a small intercept, but still the measured concentration is underestimated by the modeled values.

## 4 Summary and conclusions

Emission fluxes of secondary sulfate and dust aerosol were identified and their transport contribution was quantified based on a dataset including measurements from 16 cities in Europe and Asia. In the secondary sulfate case, 14 out of the 16 cities were used, as it was only in those that a secondary sulfate aerosol species was identified through PMF analysis. In the dust aerosol case, six cities were used as in the rest of the cities, based on PSCF analysis, dust aerosol was considered to be of local origin. There was one city whose results were not used at all (Ankara) and one city whose results were used only for dust aerosol (Lisbon). Data from Chisinau, Skopje, Banja Luka, Sofia, Belgrade, Nikšić, Kurchatov, Dushanbe, and Vilnius were only used for the secondary sulfate aerosol case.

For secondary sulfate, in the case that data from 14 stations were incorporated, the highest emission fluxes for Europe were found to be in Poland, eastern Europe, and the central and western Balkans. In Asia, the NE area of the Caspian Sea had the maximum emission flux. Its value was as high as $10 \times 10^{-12}$ kg m$^{-2}$ s$^{-1}$.

The produced emission flux solutions for secondary sulfate are evaluated by comparison to existing emission maps. The hotspots indicated by the Tikhonov regularization method appear to have high emission fluxes for OMI-HTAP and ECLIPSE SO$_2$ inventories. The Tikhonov regularization solutions for secondary sulfate do not cover the multiple significant source areas depicted in emission inventories. This probably relates to the fact that we do not have enough information with the stations and measurements at hand so as to have a high-resolution result. However, we expect that hotspot areas in the Tikhonov regularization solution are the main areas whose emissions influence the cities in the study.

When the secondary sulfate regularization solution for European cities excluding Vilnius was applied (data from 11 cities, we excluded Vilnius, Dushanbe, and Kurchatov) to aerosol masses originating from Vilnius, a relatively good agreement was found between the modeled and the measured values. This indicates the robustness of the method, as we can acquire a useful approximation to the concentration of any station for an aerosol species that is mainly transported, based only on measurements conducted in the greater geographic area. That holds even for secondary sulfate, an aerosol component that is not emitted as such but is produced in the atmosphere from precursor gases several hours after their release.

The main source area of dust aerosol for Athens, Belgrade, Debrecen, Lisbon, Tirana, and Zagreb was NW Africa (Sahara dust). There was also evident contribution from NE Africa, but it was significantly lower. The maximum emission flux was as high as $17.6 \times 10^{-12}$ kg m$^{-2}$ s$^{-1}$.

The result by the Tikhonov regularization for dust indicates NW Africa as the most significant source area, while the PSCF results for dust (Fig. 7) demonstrate a high probability for NE Africa to be a source area too. We consider that the Tikhonov regularization result is more reliable since wind speed is expected to be higher in NW Africa, and therefore more dust aerosol will be picked up by air masses there.

An overall good agreement between the measured and modeled concentrations for participating cities is not achieved. It should be noted that the result for dust is better than the result for secondary sulfate, as it has a much smaller intercept and higher coefficient of determination ($R^2$) (Figs. 5 and 9). This is probably due to the fact that the secondary sulfate concentration also depends on atmospheric chemistry.

The purpose of the development of the new method was to contribute to the air quality management process. With this new method, an improved identification of source areas for the long-range transported aerosol in comparison to PSCF analysis is achieved. Also, the relative importance of emission fluxes from each geographic grid cell is classified. This classification could be compared to existing emission inventories, resulting in possible improvements in the emission flux calculation algorithms. The new method also provides an estimate of the magnitude of emission fluxes from each grid cell. For secondary sulfate, around 60 % of the measured concentrations magnitude could be reconstructed (Fig. A8a) based on the deducted emission fluxes, while for dust, approximately 45 % could be reconstructed (Fig. A8b). This indicates that in this case, the new method significantly underestimates emission fluxes and measured concentrations. It is important to keep in mind though that if data with lower uncertainty are used, the underestimation would be significantly lower. Also, additional a priori information could lead to better performance of the method. Since we identify the pollutant source area and its relative magnitude and acquire an estimate of the measured concentrations, we can implement targeted mitigation measures. This approach can be used for any pollutant that can be simulated in FLEXPART or any similar model, without the need for an emission inventory. Ideally, we would like to implement the new method in combination with chemical transport models so as to improve

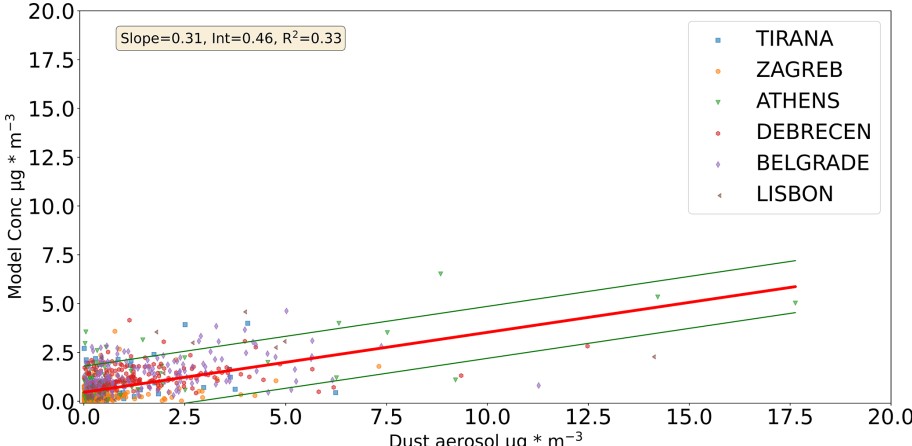

**Figure 9.** Comparison between the measured dust concentration and the modeled values based on the Tikhonov regularization solution for Athens, Belgrade, Debrecen, Lisbon, Tirana, and Zagreb. Regression line ±2 standard deviations is depicted. The measurement location is provided in the legend.

mitigation measures impact estimation. We should keep in mind that the emission fluxes deducted by the new method are averages over a period of 3 years. Emission fluxes have seasonal, monthly, weekday, and daily variations in each geographic grid cell. Therefore, the emission fluxes result derived by Tikhonov regularization can only roughly approximate the concentrations measured at the cities participating in the study. Nevertheless, we still have enough information to plan mitigation measures.

Further work could include the application of the new method on other aerosol components, like black carbon, so as to estimate its emission fluxes from each geographic grid cell.

## Appendix A: Results

### A1    Secondary sulfate PSCF: cities not included in Fig. 2

Dushanbe indicates the area east of the Caspian Sea (NE and SE) as a source. Belgrade secondary sulfate mainly stems from the Eastern Mediterranean. Chisinau indicates the southern Balkans and Poland as a source. It also indicates the area in the northeast of the Caspian Sea. Debrecen secondary sulfate also stems from the Eastern Mediterranean.

We only have 50 measurements from Nikšić, and its PSCF results are not considered statistically significant. From Sofia we only have 50 measurements, clearly not enough for PSCF analysis. Tirana indicates a transport path from Ukraine and central Europe. Kurchatov indicates secondary sulfate source areas in Siberia, probably related to gas flaring.

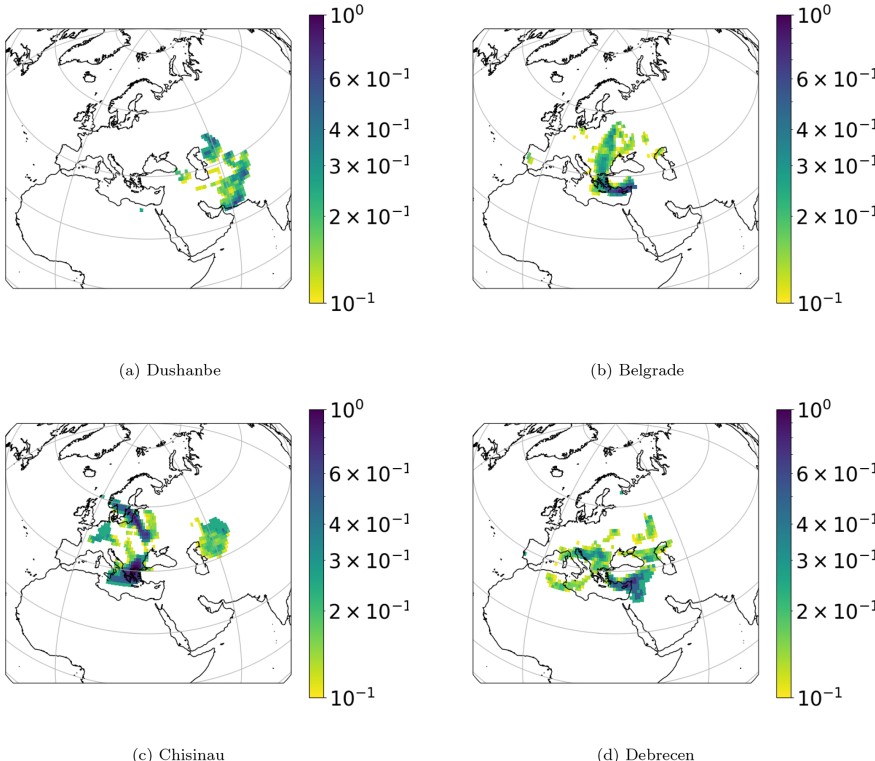

**Figure A1.** PSCF analysis for secondary sulfate aerosol for concentrations higher than the 90th percentile: Dushanbe, Belgrade, Chisinau, and Debrecen. The vertical bar represents the PSCF ratio ($\text{PSCF}_{i,j} = \text{weight}_{i,j} \times m_{i,j}/n_{i,j}$).

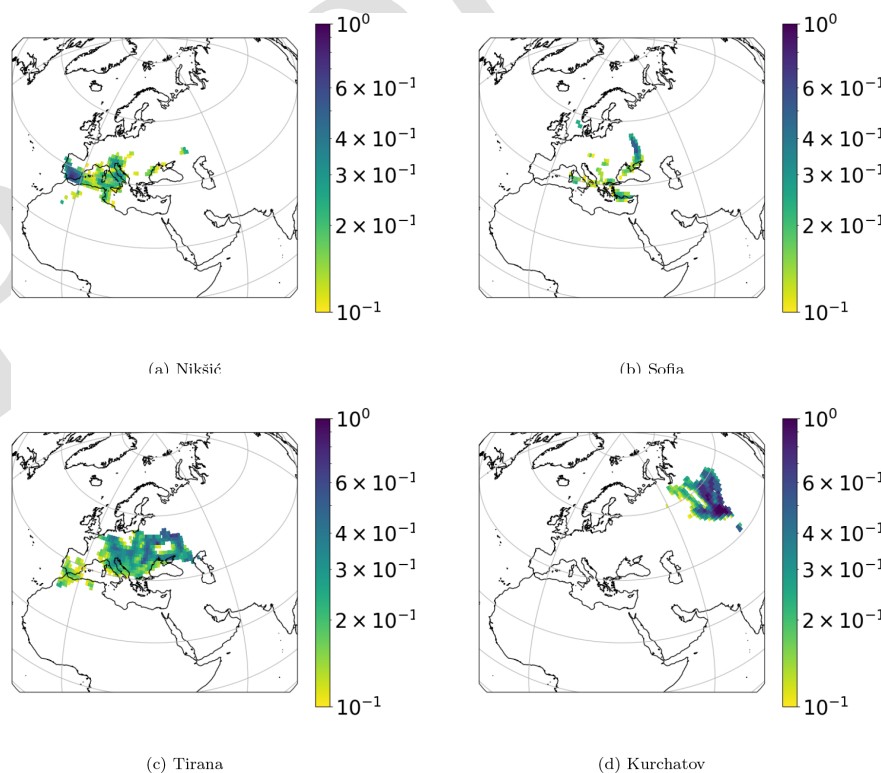

**Figure A2.** PSCF analysis for secondary sulfate aerosol for concentrations higher than the 90th percentile: Nikšić, Sofia, Tirana, and Kurchatov. The vertical bar represents the PSCF ratio ($\text{PSCF}_{i,j} = \text{weight}_{i,j} \times m_{i,j}/n_{i,j}$).

## A2   Footprint Tirana

In Fig. A3 the residence time in each grid cell (sensitivity of measurement station to emissions from each grid cell) for a height of up to 500 m is displayed.

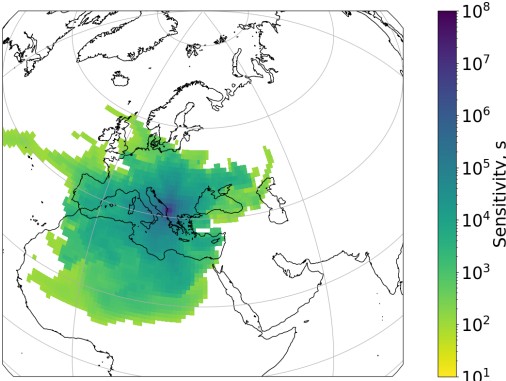

**Figure A3.** Tirana residence time in each geographic grid cell for secondary sulfate aerosol (all filter measurements). The vertical bar corresponds to seconds of residence time (sensitivity of measurement station to emissions from each grid cell). Air mass transport up to a height of 500 m a.g.l. is included.

## A3   Secondary sulfate Tikhonov regularization solutions for summer–winter: 14 cities

We observe in Figs. A4 and A5 that secondary sulfate aerosol in Dushanbe has significantly higher values in the winter, indicating influence from domestic heating. In Fig. A4 we also observe that the hotspot over Poland is reduced in summer. The other source areas indicate similar values in winter and summer.

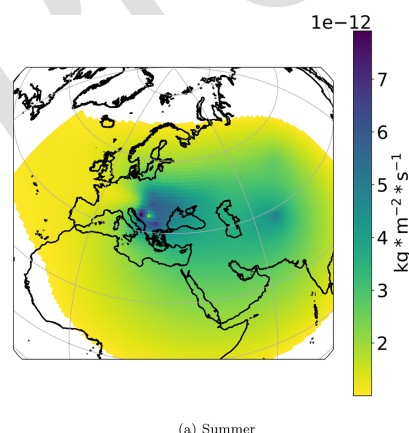
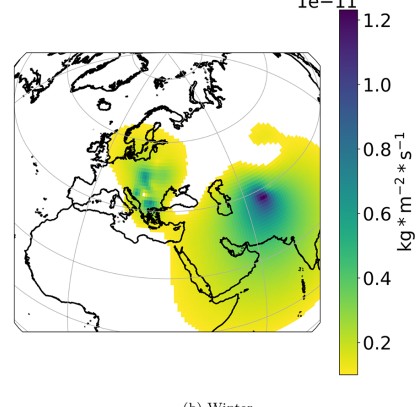

(a) Summer                                      (b) Winter

**Figure A4.** Secondary sulfate Tikhonov regularization solution for 14 cities (emission fluxes) for summer (**a**, April to September) and winter (**b**, October to March) months. The vertical bar represents the emission fluxes from each geographic grid cell in $kg\,m^{-2}\,s^{-1}$.

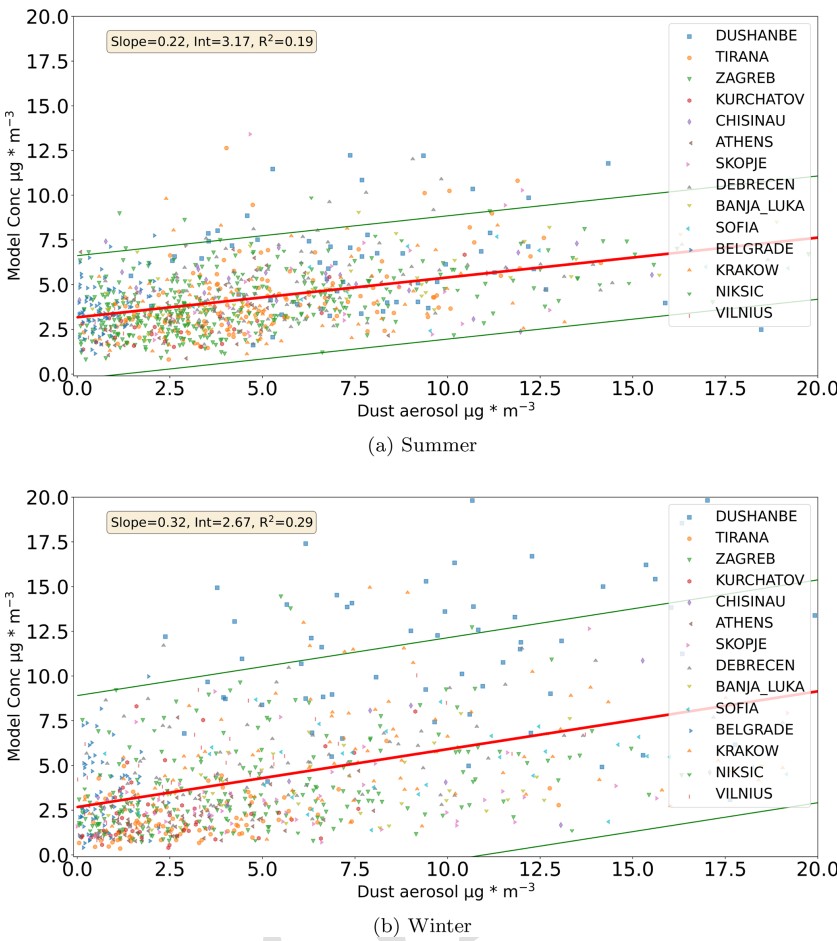

(a) Summer

(b) Winter

**Figure A5.** Comparison between modeled and measured secondary sulfate aerosol concentration in summer (April to September) and winter (October to March) months. Regression line ±2 standard deviations is depicted. The measurement location is presented in the legend. TS2

## A4   Dust PSCF: cities not included in Fig. 7

In Figs. A6 and A7 we observe that the PSCF analysis at the 90th percentile does not indicate high-emission areas for dust aerosol.

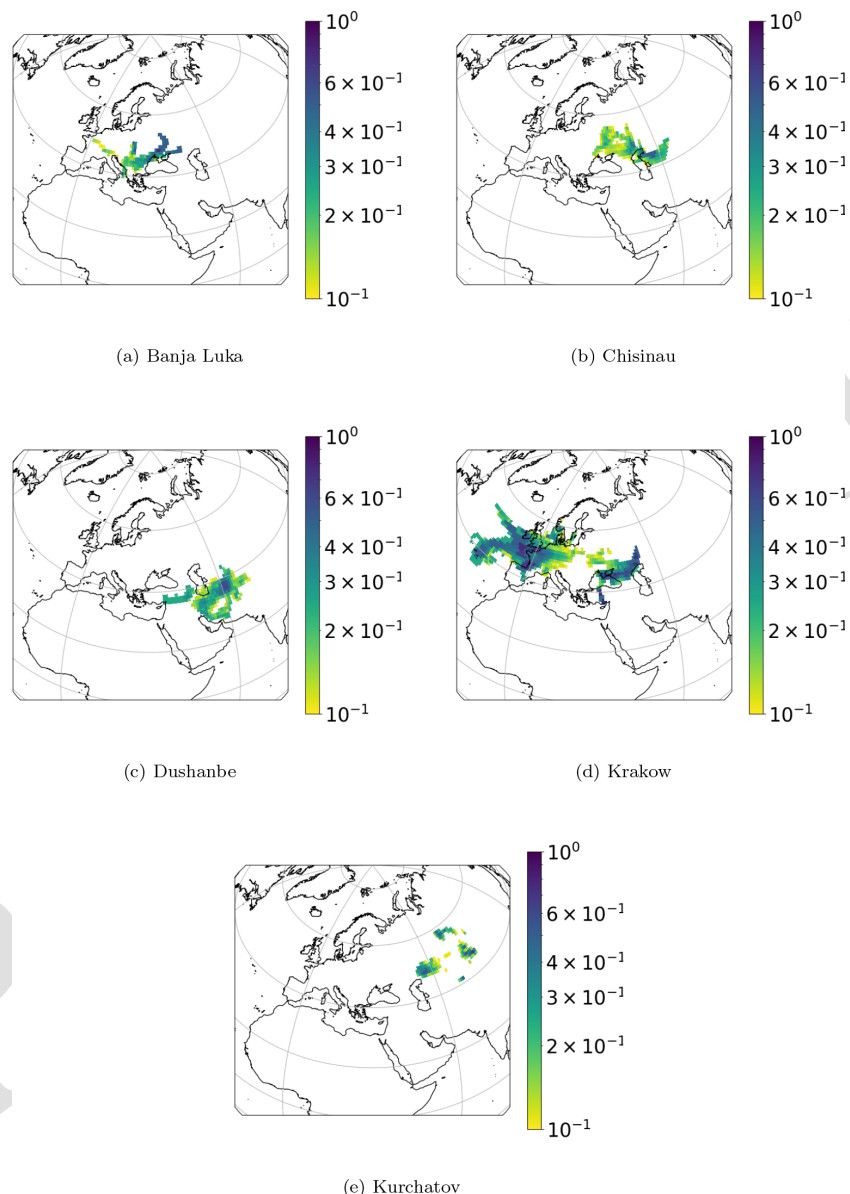

**Figure A6.** PSCF for concentrations higher than the 90th percentile for dust aerosol and for cities not included in Fig. 7. The vertical bar represents the PSCF ratio ($\mathrm{PSCF}_{i,j} = \mathrm{weight}_{i,j} \times m_{i,j}/n_{i,j}$).

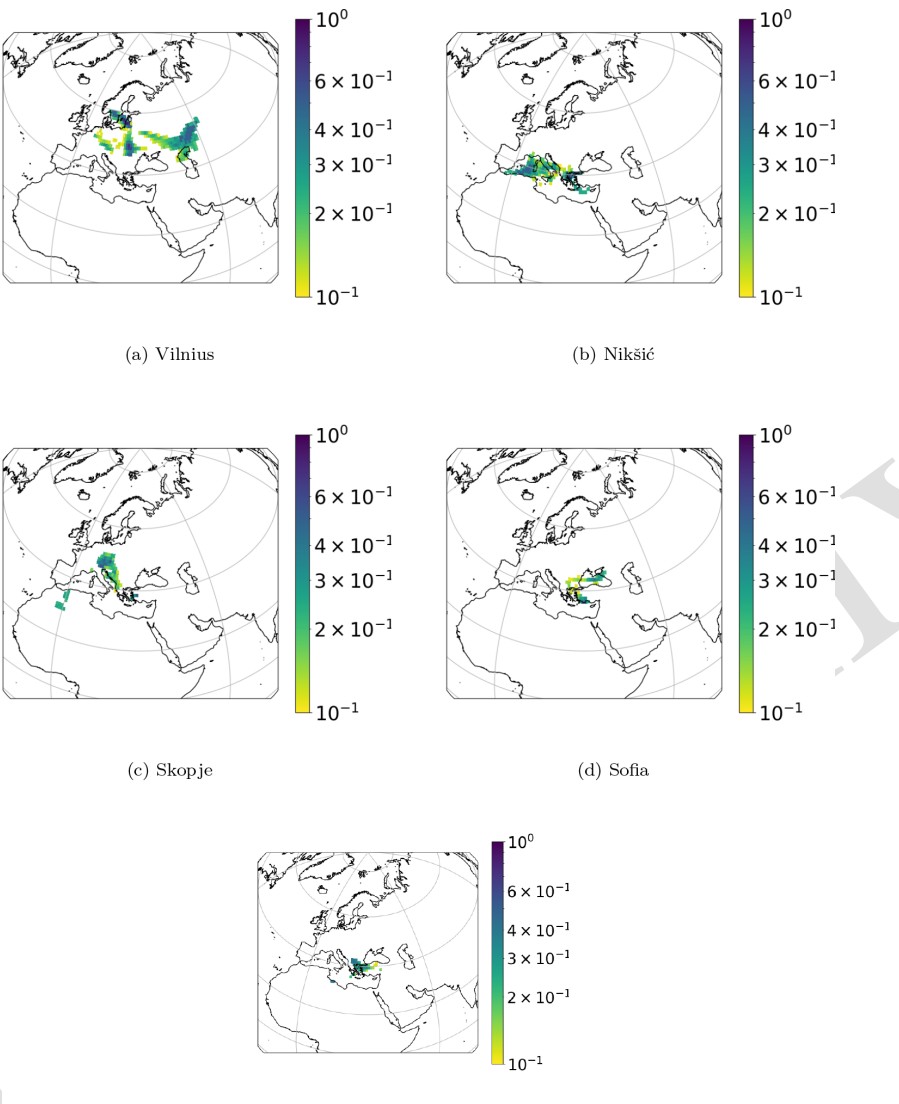

**Figure A7.** PSCF for concentrations higher than the 90th percentile for dust aerosol and for cities not included in Fig. 7. The vertical bar represents the PSCF ratio ($\mathrm{PSCF}_{i,j} = \mathrm{weight}_{i,j} \times m_{i,j}/n_{i,j}$).

## A5   Comparison between measured and modeled values for secondary sulfate and dust aerosol: no intercept

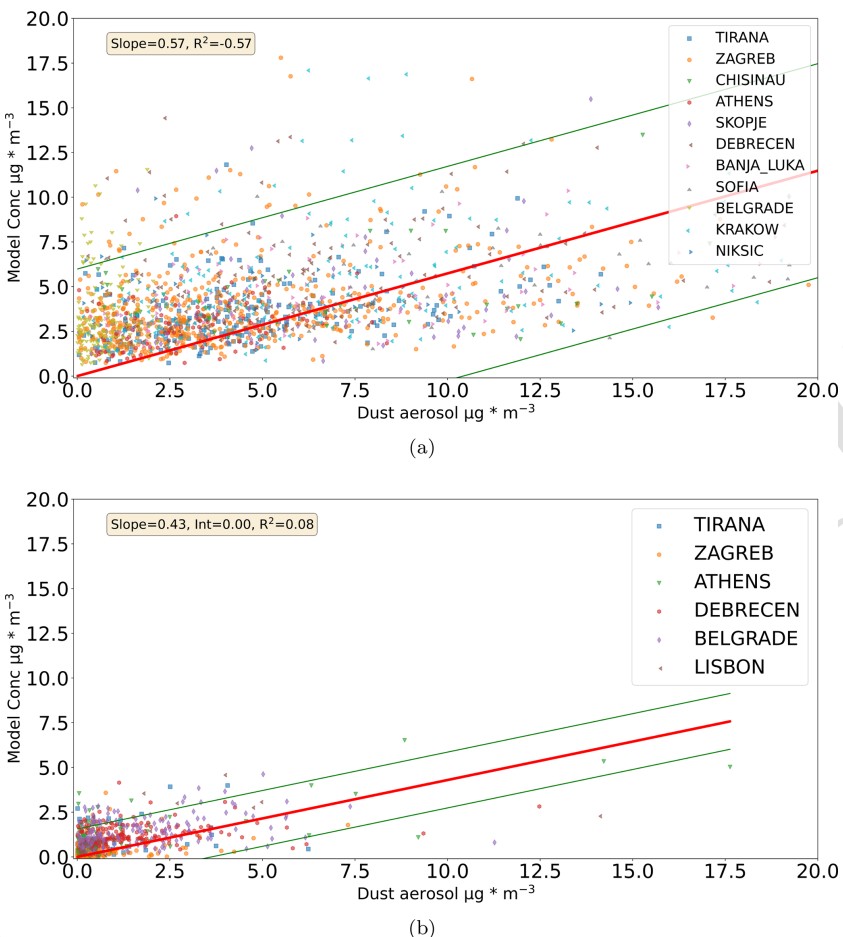

(a)

(b)

**Figure A8.** Comparison between measured and modeled values based on Tikhonov regularization solutions (no intercept). The solution for secondary sulfate using data from 14 cities is utilized in **(a)**, while in **(b)** the dust aerosol emission fluxes for the Tikhonov regularization solution is used. The regression line ±2 standard deviations is also presented. The measurement location is presented in the legend. TS3

**Code availability.**   Code will be available upon request.

**Data availability.**   The data files used to produce the results are deposited on Zenodo: https://doi.org/10.5281/zenodo.7912793 (Vratolis et al., 2023).

**Author contributions.**   SV analyzed data, interpreted results, prepared the figures, and wrote the text of the paper. ED, MIM, SMA, IB, ZK, and LS provided the PMF analysis data and contributed to interpreting results. KE contributed to interpreting results. All authors reviewed the final paper.

**Competing interests.**   The contact author has declared that none of the authors has any competing interests.

**Disclaimer.**   Publisher's note: Copernicus Publications remains neutral with regard to jurisdictional claims in published maps and institutional affiliations.

**Acknowledgements.**   This research has been supported by the program "RER/1/015 – Apportioning air pollution sources on a regional scale", 2016–2017. We acknowledge support from the Zenodo repository for hosting the data files used in the paper. We also acknowledge support from the European Centre for Medium-Range Weather Forecasts (ECMWF) for providing the meteorological data

used in the paper. We thank FLEXPART developers for providing the model and tools so as to acquire the results in the paper. We thank the developers of the Python, Numpy, Pandas, Scipy, Matplotlib, netCDF4, and Cartopy software packages. Finally, we would like to thank Nikolaos Evangeliou, Delia Arnold, and Ignacio Pisso for their help in setting up the FLEXPART model.

**Financial support.** This research has been supported by the International Atomic Energy Agency (grant no. RER/1/015).

**Review statement.** This paper was edited by Barbara Ervens and reviewed by two anonymous referees.

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

## Remarks from the typesetter

**TS1**    We noticed that the new figure includes considerably more data than the current version. Please note that substantial changes like this are not possible at this stage. In case these changes are essential, we would have to forward your requests to the handling editor for approval. To explain the corrections needed to the editor, please send me the reason why these corrections are necessary. Please note that the status of your paper will be changed to "Post-review adjustments" until the editor has made their decision. We will keep you informed via email.