# Peer review of "A new method for the quantification of ambient particulate matter emission fluxes"

_Atmospheric Chemistry and Physics, 2022_

## Author Comment (AC1)

The idea of the paper of obtaining fluxes of emission for dust and sulphate is very interesting. However, the presentation is poor and the support to results and conclusions is weak in my opinion.

I am in the interphase of major revision or rejection.

*The authors are grateful for all the reviewer's comments and suggestions.*

*The corresponding author would like to state the following:*

*There was an error in the units of the emission fluxes from each grid cell.*

*We will correct this in the manuscript and the deducted emission maps.*

*Our answers on your questions follow in italics:*

1. Change in tittle and all text 'emission factors' by 'emission fluxes'. 'Emission factor' in 'emission and projections' has a very well-defined meaning, with kg or t/unit of activity. When referring to emission or deposition per area and time, the term 'fluxes' is used.

*We will replace in the title and all text 'emission factors' by 'emission fluxes'. The title will now be: 'A new method for the quantification of ambient particulate matter emission fluxes'*

2. References for health studies are ok but old, please update at least with the most recent papers on the Global Burden of disease and 2021 WHO AQ guidelines

*We will add the following articles:*

*Ghosh et al., 2021, Ambient and household $PM_{2.5}$ pollution and adverse perinatal outcomes: A meta-regression and analysis of attributable global burden for 204 countries and territories, PLOS Medicine, 18, e1003 718;*

*Burkart et al.,Estimates, trends, and drivers of the global burden of type 2 diabetes attributable to $PM_{2.5}$ air pollution, 1990–2019: an analysis of data from the Global Burden of Disease Study 2019, The Lancet Planetary Health, 6, 2022;*

*Pandey et al., 2021; Health and economic impact of air pollution in the states of India: the Global Burden of Disease Study 2019, The Lancet Planetary Health, 5, e25–e38*

*WHO, 2021.WHO global air quality guidelines. Particulate matter ($PM_{2.5}$ and $PM_{10}$ ), ozone, nitrogen dioxide, sulfur dioxide and carbon monoxide, Tech. rep., World Health Organization, 2021.*

3. Very repetitive the paragraph below. Try to send the messages only once and add references in all cases:

*We will replace the paragraph with the following:*

*'In order to identify and quantify aerosol sources and corresponding source areas, significant effort is required by the Scientific Community. When this information is acquired, measures can be applied so as to improve air quality. Source apportionment methods are widely used for air quality*

*management, by providing information on the relationship between air pollutant sources and their concentrations. The quantification of the sources of air pollution, both in terms of their sectorial and spatial origins, constitutes an essential step of the air quality management process (Wesseling et al., 2019).'*

4. Better justify that the method used can be applied for secondary PM components, such as sulphate. For dust it is clear but in different seasons the $SO_2$ oxidation velocity might change and sulphate being formed faster or slowly and then the distance to the origin might change artificially for this. At least evaluate what effect it might have.

*We will add in the manuscript in the FLEXPART description, section 2.2:*

*In the case of Secondary Sulfate, we have to keep in mind that $SO_2$ is the primary emitted species and Secondary Sulfate is produced in the atmosphere through chemical reactions in gas and liquid phase. In order to calculate the uncertainty that this error induces to the calculated footprint, we refer to residence times in the atmosphere reported by Seinfeld and Pandis (1998), page 66. The $SO_2$ mean residence time reported due to dry deposition is 60 hours, its residence time due to wet deposition is 100 hours, and its residence time due to transformation to Secondary Sulfate is 80 hours. The resulting $SO_2$ residence time due to wet and dry deposition is 37.5 hours, while if we also include the transformation to Sulfate the overall mean residence time is 25 hours. The corresponding wet and dry deposition residence time indicated for Secondary Sulfate is 80 hours. Therefore, in such a case, $SO_2$ deposits (wet and dry deposition) twice as fast as Secondary Sulfate. These calculations correspond to the mid-latitudes (45°-65° North) according to Rodhe et al. FLEXPART model is provided with a Secondary Sulfate aerosol particle size distribution and it compensates for wet and dry deposition as it follows the species backward in time. The error in the calculation of the residence time in each geographic grid cell is mainly due to not accounting for the enhanced deposition of $SO_2$ for 1-2 days just after emission. But this enhanced wet and dry deposition for $SO_2$ should be applied only for a small fraction of the travel time. The mean error in residence time due to this discrepancy is expected to be close to 10%.*

*We also have to keep in mind that we do not present emission fluxes of the $SO_2$ emissions, but the origin of Secondary Sulfate aerosol measured at each station, if it was produced as such in the emitting grid cell. Therefore we report the combined effect of $SO_2$ emissions, air mass transport and environmental conditions that produce the Secondary Sulfate aerosol measured in the stations participating in the study. That is why the authors believe that we cannot apply the fluxes derived to very distant measurement stations, whose environmental conditions might be very different from the stations in the study. Also, the estimated error is calculated based on values derived for the mid-latitudes.*

5. Not clear to me how regional from long range sulphate and dust can be distinguished.

*According to Seinfeld and Pandis (1998), page 66, Secondary Sulfate has a mean residence time of 80 hours in the atmosphere. We therefore expect that most of the Secondary Sulfate aerosol measured at each station, has been produced in the atmosphere within the previous week. This would probably correspond to regional transport. The authors expect that most of the Dust aerosol measured at each*

*station would be regional, since it has a much larger aerodynamic mean diameter (3.1 micrometers), leading to much faster deposition velocity.*

*We will add in the manuscript in the FLEXPART model description, section 2.2:*

*"Since Secondary Sulfate has a mean residence time of 80 hours in the atmosphere, as reported by Seinfeld and Pandis (1998), we expect that most of the Secondary Sulfate aerosol measured at each station, has been produced in the atmosphere within the previous week.*

*This would probably correspond to regional transport, not global. The authors expect that most of the Dust aerosol measured at each station would be regional, since it has a much larger aerodynamic mean diameter of 3.1 micrometers, leading to much faster deposition velocity. In any case, both species are followed backward in time for 20 days, and residence times are attributed for all geographic grid cells. However, we use for the inversion the residence times in each cell for the area between latitude -30° to 90° and longitude from -40° to 140°."*

6. You stated in text that 16 cities are studied and only 14 are indicated in the maps of Figure 1.

*In Figure 1a, Vilnius and Krakow are displayed but not included in the subtitle of the Figure.*

*We will correct it in the manuscript.*

7. You explain that only 14 were selected, but why 2 were excluded give reasons in methodology.

*In lines 143 to 145 we state:*

*"We did not exclude any station with an identified Secondary Sulfate source from the analysis for this species, as we consider that Secondary Sulfate and its precursor gases are emitted from many source areas in Europe and Asia (14 out of 16 stations were included)."*

*We will add in the manuscript in section 3.1 (Secondary Sulfate aerosol):*

*In 2 cities (Ankara, Lisbon), no Secondary Sulfate concentration was indicated by the PMF analysis. Therefore 14 out of the 16 cities could be included, namely Tirana, Zagreb, Chisinau, Athens, Skopje, Debrecen, Banja-Luka, Sofia, Belgrade, Krakow, Montenegro, Kurchatov, Dushanbe, Vilnius (around 2,050 measurements).*

*In section 2.3 on Tikhonov regularization we will add at the end:*
*A Secondary Sulfate aerosol species was identified in 14 out of 16 cities in the study, and therefore the two cities without this source (Ankara, Lisbon) were excluded. In a small number of samples in the 14 cities included in the study, negative contributions were apportioned. These samples were excluded from the dataset used in the Tikhonov regularization.*

*Dust aerosol source was identified in 16 cities. Nevertheless, after the PSCF analysis for Dust aerosol, only 6 cities were included in the Tikhonov regularization dataset. That is because the PSCF analysis indicated that most of the Dust aerosol identified was of local origin (Dust resuspension). Filter samples that had negative Dust source apportionment contributions were also excluded.*

8. For a number of cities did you applied PMF with less than 50 samples. Is this right? I do not think so.

*In lines 62-64 we state:*

*Further uncertainties to the source apportionment results are introduced by the fact that the stations of Chisinau, Sofia, Niksic, Lisbon, Ankara and Vilnius have available only 50 filter samples.*

*We will add in the manuscript in section 2.1 (PM sampling stations and filter analysis):*

*"We have not applied PMF to less than 50 samples in any of the cities. We applied PMF analysis on datasets with 50 samples from 5 cities (Chisinau, Sofia, Vilnius, Montenegro, Lisbon and Ankara). 50 samples have been recorded as the minimum necessary for a meaningful source apportionment analysis according to Johnson et al. (2001). Having said that, it has been identified in the past that small datasets (number of samples close to 50) pose an extra challenge when used for PMF because the solution is strongly affected by rotational ambiguity, and the overall uncertainty is increased. Before using the results we have fully assessed the uncertainty of the SA solution using the enhanced tool offered by EPA PMF 5.0. The uncertainty was within acceptable limits.  We included these measurements because they are valuable, as aerosol data from these areas are scarce, and also, including them would diversify the origin of air masses used in the identification of source areas and emission fluxes, making our results more precise."*

9. R143-147 you select and exclude sites without supporting reasons.

*In lines 158-160 we state for the Secondary Sulfate aerosol result for Zagreb:*

*"We included the emission results for Zagreb as it was the station with the largest number of filter samples that participated in the study. This case indicates the results we could expect when we use data from a single station."*

*In lines 183-184 we state:*

*"Vilnius station was chosen for the demonstration of the results as it is situated on the edge of the area that the rest of the European stations of the study cover."*

*We will include in the manuscript (section 3.1,  Secondary sulfate aerosol):*

*Our first approach was to apply the Tikhonov regularization to data from the 6 cities indicated by the PSCF analysis in Figure 2. Then we applied regularization to all 14 cities. We applied the Tikhonov regularization to all 14 cities with an identified Secondary Sulfate concentration, as we consider that Secondary Sulfate and its precursor gases are emitted from many source areas in Europe and Asia and we needed as many stations and measurements as possible in order to identify them.*

*We also produced two more emission flux results: One including only measurements from Zagreb (around 600 measurements), and one including all participating European cities except Vilnius (around 1,800 measurements). The first result is indicative of using a dataset from just one measurement station. In the second result we exclude Vilnius, Dushanbe and Kurchatov data. Dushanbe and Kurchatov are situated in a significant distance from other stations, out of the region of Europe. Vilnius on the other hand is on the edge of the area that is covered by European stations.*

*This result was produced as we wish to evaluate if by its use we could predict Secondary Sulfate concentration in Vilnius.*

10. R161 cite Figure 3.

*We will add in the manuscript the OMI – HTAP emission map provided by NASA (https://avdc.gsfc.nasa.gov/pub/data/project/OMI_HTAP_emis/v1.1) (Liu et al, 2018).*

*We will add in the manuscript in section 3.1, Secondary Sulfate aerosol:*

*We also observe in Figure 3 that the areas indicated by the Tikhonov regularization solution for 14 cities (Figure 3b), namely the Central and Western Balkans, South Poland and the area East of the Caspian Sea, are apparent also in the ECLIPSE database map (Klimont et al., 2017).*

*When we compare the OMI – HTAP emission map for $SO_2$ and the emission map acquired by the Tikhonov regularization for the investigated European cities excluding Vilnius (Figure 4a), we observe many similarities.*

11. All this section on sulphate is very confuse, you select some cities, then you do not include because low samples and then use others. At the end the reader does not know what you have done and why you exclude and select ones or the others.

*We will add in the methodology on Tikhonov regularization the datasets used for Secondary Sulfate and Dust aerosol and some justification (please refer to point 7). We will also add justification on the selection of cities at the beginning of the Secondary Sulfate and Dust results sections (please see point 9).*

*The authors believe that this is now addressed in the manuscript.*

*We will also add in the manuscript, at the end of section 1 (Introduction):*

*We have to mention here that the emission fluxes retrieved are subject to air mass transport paths, atmospheric conditions and atmospheric chemistry. In other words, if a geographic grid cell emits a pollutant, but air mass transport does not allow these emissions to reach any of the measurement stations in the study, this cell will not be attributed the emission flux that it has. For species like Secondary Sulfate, identical precursor gases emission fluxes could lead to different aerosol concentrations, depending on atmospheric conditions and chemistry. It is also possible that locally produced aerosol (that is within the station grid cell), cannot be correctly associated to residence time in the grid cell. That is because emission fluxes in the vicinity of the measurement stations have a very small residence time until they arrive to the station and a very high impact on the measured concentration. Despite of these potential problems, the information on specific geographic grid cells that actually impact the measurement stations area is focused on where we have to apply mitigation measures for long range transport.*

*We will also add in the Tikhonov regularization description, section 2.3:*

*Sulfate or Dust concentration at each measurement station is due to local production and long range transport.With Tikhonov regularization we aim not to perfectly reconstruct the concentrations*

measured at each station, but find $x_{exact}$. For local aerosol, produced in the vicinity of the measurement station, we expect that we will have a high impact on the concentration and a small Impact on the residence time in the station grid cell. Therefore, matrix A mostly corresponds to the part of the concentration that is transported to the site from other grid cells and cannot accurately describe local aerosol. However, we expect that during the Tikhonov regularization procedure, while we search for $x_{exact}$, and we depart from perfectly reconstructing measured concentration, local aerosol will be attributed as noise and we will recover the correct emission fluxes.

12. Why you did not use OMI or TROPOMI for SO2 concentrations in addition to Eclipse maps?

These show much better the SO2 hotspots.

*We have added in the manuscript the 1° \* 1° degree OMI_HTAP emissions map (Liu et al, 2018) for 2015.*

13. Furthermore, I do not see properly that Eclipse and your maps show similar high SO$_2$ regions.

*The most significant areas that appear to affect the measurement stations in the European region are in the Balkans and South Poland. We have added patches so as to indicate these areas in the OMI_HTAP maps included now in the manuscript. The same areas are apparent also in the ECLIPSE database emission map.*

14. Figure 4 you do not reach a good agreement for Vilnius but you do not mention for the other cities.

*We display Vilnius because it was not included in the dataset (along with Kurchatov and Dushanbe) that produced the solution in Figure 3b. The aim was to show that we could produce a meaningful approximation to the concentration in a nearby city whose data were not used in the procedure to acquire the solution. We should have stressed that probably locally produced Secondary Sulfate (produced within the same grid cell as the station) is included in the concentration and this value cannot be represented correctly in the solution, which is analogous to residence time in the grid cells.*

*We will add in the manuscript the least square regression for each Secondary Sulfate and Dust solution between the measured and modeled concentrations (based on the Tikhonov regularization solution for each case).*

*We will also add the following in the manuscript, in section 4 (Summary and Conclusions):*

*"The results display that we do not have a good agreement between the measured and modeled concentrations, which probably indicates that resuspended Dust and locally produced Secondary Sulfate is present in the measured concentrations. This was expected, as with Tikhonov regularization we depart from perfectly reproducing the measured concentrations (fidelity term). We have to mention here that the purpose of the regularization method is to identify and quantify emission fluxes from each geographic region. In this process, the part of the concentration that can be represented is mainly the one corresponding to long range transport."*

15. You state that it might be precipitation the reason for the lack of this agreement but no support is given for this.

*We state in lines 182-183:*

*From July to September are the months with the most precipitation in Vilnius (WMO, 2013).*

*We will remove this statement.*

*We will add in section 3.1.1 (The case of Vilnius):*

*The lowest PM2.5 concentrations in the dataset are observed during August, September and nearly the end of October. This could also be related to the beginning of the winter season, with increased emissions due to heating.*

16. I am not able to identify in the result section on sulphate how and what are the emission fluxes you cited in the abstract.

*We will add in section 3.1, Secondary Sulfate aerosol:*

*In Figure 3b which corresponds to the solution for all available data (14 stations), the highest values are as follows:*
*For the area East of the Caspian Sea, the maximum value is $10*10^{-12}$ $kg*m^{-2}*s^{-1}$ for latitude 37°-38° North and longitude 67°-68° East. In the OMI-HTAP map, the maximum value in the area is in latitude 39°-40° North and 65°-66° East, with a value of $7.7*10^{-8}$ $kg*m^{-2}*s^{-1}$.*
*For the area in the west Balkans, the maximum value is $7.8*10^{-12}$ $kg*m^{-2}*s^{-1}$ for latitude 44°-45° degrees North and longitude 16°-17° East. In the OMI-HTAP map, the maximum value in the area is in latitude 44°-45° North and 18°-19° East, with a value of $9.2*10^{-8}$ $kg*m^{-2}*s^{-1}$.*
*For the area in Poland, the maximum value is $6.1*10^{-12}$ $kg*m^{-2}*s^{-1}$ for latitude 49°-50° degrees North and longitude 19°-20° East. In the OMI-HTAP map, the maximum value in the area is in latitude 51°-52° North and 19°-20°ast, with a value of $5.3*10^{-8}$ $kg*m^{-2}*s^{-1}$.*
*For the area in the Central Balkans, the maximum value is $8.3*10^{-12}$ $kg*m^{-2}*s^{-1}$ for latitude 42°-43° North and longitude 20°-21° East. In the OMI-HTAP map, the maximum value in the area is in latitude 44°-45° North and 18°-19° East, with a value of $9.3*10^{-8}$ $kg*m^{-2}*s^{-1}$.*

17. Dust I am not able to see in the map that NW Africa is the main source, especially for Tirana.

*We will add in the manuscript in section 3.2 (Dust aerosol):*

*In Figure 7 the PSCF for the 90th percentile for Dust aerosol is presented. For Tirana two paths can be seen: In the first path, winds from the Atlantic Ocean pass over NW Africa, then the Mediterranean Sea and subsequently reaching Tirana. In the second path winds from the Atlantic Ocean pass over NW Africa, then NE Africa and the Mediterranean Sea, subsequently reaching Tirana. The second path is by far the prevailing one for the 90th percentile highest concentrations of Dust aerosol for Tirana, as can be seen in Figure A.3 in the appendix. Please keep in mind that the residence times depicted correspond to a height up to 500 m so as to always be within the boundary layer. Therefore, while the Dust load could be mainly picked up in both cases in NW Africa, due to longer residence times in NE Africa, this area could appear as the most probable to be the one that emits Dust aerosol. This could be partly due to the fact that as the air masses travel over Africa at low altitude, wind speed is reducing due to higher friction over land in comparison to when they travel over the Sea (Atlantic or Mediterranean). The air masses probably have higher speed over NW Africa and this results in more dust being picked up in this area. Some Dust aerosol could be picked up from NE*

*Africa and its origin could also be the Arabian Peninsula. This path is also evident in Figure 8, where a weak emission area is indicated in the NE Africa. While for the PSCF analysis, Tirana, Zagreb and Belgrade indicate high probability for NE Africa to be a source area, this is not the case for the Tikhonov regularization result. In Figure 8, the result indicates that NW Africa is by far the most significant Dust aerosol source area for the 6 cities (Athens, Belgrade, Debrecen, Lisbon, Tirana and Zagreb) whose data were used. NE Africa also has a hotspot in Figure 8, but its contribution was significantly lower when the data from these 6 stations are combined.*

18. IN many cases is N Africa, but not the well known large sources from NW Africa, Central and S Argelia, Mauritania, Sahara,....

That are not covered by the map patches.

*We will add in the manuscript in section 3.2 (Dust aerosol):*

*In the borders between Mauritania, Algeria and Mali, the highest emission fluxes are identified (lat 27° N, long -4° E) which are as high as $17.6*10^{-12}$ kg$*m^{-2}*s^{-1}$.*

19. Summary: 16 or 14 cities?

*The initial dataset included 16 cities. A Secondary Sulfate aerosol source was apportioned through PMF in 14 cities. We used all 14 of them. We used for Dust aerosol results from 6 cities. There was one city whose results were not used (Ankara) and one city whose results were used only for Dust (Lisbon).*

*We will add in the manuscript in the Summary section (Summary and Conclusions):*

*Emission fluxes of Secondary Sulfate and Dust aerosol were identified and their transport contribution was quantified based on a dataset including measurements in 16 cities in Europe and Asia. In the Secondary Sulfate case 14 out of the 16 cities were used as only in those a Secondary Sulfate aerosol source was apportioned through PMF analysis. In the Dust aerosol case 6 cities were used as in the rest of the cities, based on PSCF analysis, Dust aerosol was considered to be of local origin. There was one city whose results were not used at all (Ankara) and one city whose results were used only for Dust aerosol (Lisbon). Data from Chisinau, Skopje, Banja-Luka, Sofia, Belgrade, Montenegro, Kurchatov, Dushanbe, Vilnius were only used for the Secondary Sulfate aerosol case.*

*In section 2.1 (PM sampling stations and filter analysis) we will add in the beginning:*

*More than 2,200 PM2.5 samples were collected in urban and sub-urban background stations from 16 European and Central Asian cities (Tirana, Zagreb, Chisinau, Athens, Skopje, Debrecen, Banja-Luka, Sofia, Belgrade, Krakow, Montenegro, Kurchatov, Dushanbe, Vilnius, Lisbon, Ankara)*

20. Summary you give the quantitative emission fluxes for both dust and sulphate without showing results on it in the prior sections?????

*We will add in the Results section the emission fluxes for Dust and Secondary Sulfate, as mentioned in points 16 and 18.*

*References:*

Johnson, T., Guttikunda, S., Wells, G., Artaxo, P., Bond, T., Russell, A., Watson, J., and West, J.: Tools for Improving Air Quality Management: A Review of Top-Down Source Apportionment Techniques and Their Application in Developing Countries, World Bank, Washington, DC, 2011.

Liu, F., Choi, S., Li, C., Fioletov, V. E., McLinden, C. A., Joiner, J., Krotkov, N. A., Bian, H., Janssens-Maenhout, G., Darmenov, A. S., and da Silva, A. M.: A new global anthropogenic SO2 emission inventory for the last decade: a mosaic of satellite-derived and bottom-up emissions, Atmospheric Chemistry and Physics, 18, 16 571–16 586, https://doi.org/10.5194/acp-18-16571-2018, 2018.

Rodhe, H.: Budgets and turn-over times of atmospheric sulfur compounds, Atmospheric Environment (1967), 12, 671–680, 370 https://doi.org/10.1016/0004-6981(78)90247-0, 1978.

---

## Author Comment (AC2)

General comments

The paper "A new method for the quantification of ambient particulate matter emissions" by Vratolis et al. investigates an inverse method based on receptor and back trajectory models to identify and quantify the distribution and intensity of dust and sulfate emission sources.

The paper is well written, with concise and clear statements, and it does not require any substantial review of syntax and language.

The paper content fits the scope of ACP, however at this stage is not suitable for publication due to several major issues that need to be addressed.

*The authors are grateful for all the reviewer's comments and suggestions.*

*The corresponding author would like to state the following:*

*There was an error in the units of the emission fluxes from each grid cell.*

*We will correct this in the manuscript and the deducted emission maps.*

*Our answers on your questions follow in italics:*

In particular:

1. The authors introduce an optimization function they apply to define the matrix of emission density (called "x"), but the discussion about the choice and the robustness of the solution is totally missing

*A very similar optimization function has been used in Stohl et al, (2009) paper with the title: "An analytical inversion method for determining regional and global emissions of greenhouse gases: Sensitivity studies and application to halocarbons".*

*The robustness of the method is displayed when we examine Figures 3a and 3b: When we use data from Kurhatov, Dushanbe and Vilnius stations, the result for the European area is almost identical to the result when we include all cities. The only difference is the area East of the Caspian Sea, which is quite distant from the cities in Europe. Since the residence time of Secondary Sulfate in the atmosphere according to Seinfeld and Pandis (1998) is 80 hours and usually the prevailing winds are westerlies, the European cities in the study cannot identify these emissions. In Figure 3c where the*

*result for Zagreb is displayed, similarities to Figure 3b are apparent (area in NE Europe, West and Central Balkans), but due to the fact that we have less measurements and only one station the result lacks specificity.*

*We will also add the Tikhonov regularization emission fluxes maps for Secondary Sulfate for 14 cities during winter and summer. In summer the hotspot east of the Caspian Sea almost disappears, indating that these emissions probably relate to heating. In South Poland the hotspot is significantly reduced. Hotspots on Western and Central Balkans appear to have similar values in winter and summer, indicating that they possibly originate from power plants.*

*We will also add the comparison between Tikhonov regularization modeled Secondary Sulfate concentrations in relation to measured ones.*

2. The optimization formula seems reasonable for non-reactive pollutants, but it is not clear if the simple approach A*x = b can be applied also for reactive pollutants like SO$_2$ producing sulphate through chemical reactions

*In this method we follow the Secondary Sulfate aerosol species concentration at each station backward in time and try to locate the area that it originated from. In the mid latitudes, Seinfeld and Pandis (1998), page 66, report a residence time for Secondary Sulfate of 80 hours and a residence time for SO$_2$ of 25 hours. That means that our approach is on average correct for most of the travel time of the emitted Sulfur that is within the SO$_2$ at the beginning and Secondary Sulfate aerosol later.*

*We will add in the manuscript in the FLEXPART description, line 83:*

*In the case of Secondary Sulfate, we have to keep in mind that SO$_2$ is the primary emitted species and Secondary Sulfate is produced in the atmosphere through chemical reactions in gas and liquid phase. In order to calculate the uncertainty that this error induces to the calculated footprint, we refer to residence times in the atmosphere reported by Seinfeld and Pandis (1998), page 66. The SO$_2$ mean residence time reported due to dry deposition is 60 hours, its residence time due to wet deposition is 100 hours, and its residence time due to transformation to Secondary Sulfate is 80 hours. The resulting SO$_2$ residence time due to wet and dry deposition is 37.5 hours, while if we also include the transformation to Sulfate the overall mean residence time is 25 hours. The corresponding wet and dry deposition residence time indicated for Secondary Sulfate is 80 hours. Therefore, in such a case, SO$_2$ deposits (wet and dry deposition) twice as fast as Secondary Sulfate. These calculations correspond to the mid-latitudes (45°-65° North) according to Rodhe et al. FLEXPART model is provided with a Secondary Sulfate aerosol particle size distribution and it compensates for wet and dry deposition as it follows the species backward in time. The error in the calculation of the residence time in each geographic grid cell is mainly due to not accounting for the enhanced deposition of SO$_2$ for 1-2 days just after emission. But this enhanced wet and dry deposition for SO$_2$ should be applied only for a small fraction of the travel time. The mean error in residence time due to this discrepancy is expected to be close to 10%.*

*We also have to keep in mind that we do not present emission fluxes of the SO$_2$ emissions, but the origin of Secondary Sulfate aerosol measured at each station, if it was produced as such in the emitting grid cell. Therefore we report the combined effect of SO$_2$ emissions, air mass transport and environmental conditions that produce the Secondary Sulfate aerosol measured in the stations participating in the study. That is why the authors believe that we cannot apply the fluxes derived to*

*very distant measurement stations, whose environmental conditions might be very different from the stations in the study. Also, the estimated error is calculated based on values derived for the mid-latitudes.*

3. The validation of the obtained results is limited to one species and one site, while it should be extended to both pollutants and a significant number of sites

*Sulfate or Dust concentration at each measurement station is due to local production and long range transport. With Tikhonov regularization we aim not to perfectly reconstruct the concentrations measured at each station, but find x in the equation b = A x, where b is the concentration measured at each station (local and long range transport aerosol), A is the residence time matrix for each grid cell and x the emissions map. A corresponds to the part of the concentration that is transported to the site from other grid cells. Therefore, we expect that during the Tikhonov regularization procedure, while we search for $b_{exact}$, local aerosol will be attributed as noise and we will recover the correct x (emission map). We also have the problem of different uncertainty levels at each station due to the variations to the methodologies and materials used.*

*We have plotted the regularization solution predicted to measured values for all sites participating in the study. For secondary sulfate $R^2$ is low, the slope is approximately 0.3 and the intercept is approximately 3 $\mu g/m^3$.*

*For Dust aerosol the slope is 0.3, the intercept is approximately 0.5 $\mu g/m^3$ and $R^2$ is 0.33.*

*We will add in the manuscript in the Results sections for Secondary Sulfate and Dust:*

*When we compare the result of the regularized solution for Secondary Sulfate using data from 14 cities to measured values at each station (Figure 5b), the agreement is not good. This could be partly due to the fact that air mass transport cannot account for the locally produced aerosol (i.e. Secondary Sulfate aerosol produced within the same grid cell and very close to the measurement station will have short residence time in the cell and high impact on the measured value)*

4. Beyond these major concerns, authors should also:

Better clarify the relationship *between PSCF and matrix A, in terms of both definition and use*

*We will add in the manuscript (section 2.3, Tikhonov regularization):*

*In our particular case, each row of A matrix corresponds to FLEXPART model sensitivity (residence time in each grid cell) for each filter measurement, and each column of A matrix corresponds to a specific geographic grid cell sensitivity for all filter measurements. b corresponds to the actual species mass concentration for each filter, while x is the emission flux from each geographic grid cell. In other words we try to extract information associated to residence time in each grid cell for each filter measurement.*

*We will also add in the manuscript (section 2.5, Potential source contribution function (PSCF)):*

*We apply the PSCF analysis for each measurement station and each aerosol species. The information that we use is the overall residence time for all filters in each station ($n_{ij}$) and the overall residence times in each grid cell for the filter measurements with the highest Secondary Sulfate or Dust aerosol concentrations ($m_{ij}$). In other words we extract information from the sum of residence times of all filters and the sum of residence times for filters with the highest concentration (90$^{th}$ percentile).*

Check the terminology, particularly when referring to "source" and "concentration"

*We will add in the manuscript at the end of the Introduction section:*

*From now on, we refer to "source apportioned concentration by PMF" as "concentration" and to "geographic grid cell source areas emission fluxes" as "emission fluxes".*

There are also a several specific comments that should be addressed, that are listed below.

Specific comments and Technical corrections

P2 R48 - Does the concept of "smooth solution" require additional details?

*We will add in the Tikhonov regularization section:*

*A smooth solution is obtained when the L matrix requires that the difference between two neighboring cells is minimum. In other words, when the regularization matrix L is the first-order discrete derivative operator, it imposes smoothness on the solution (M. Donatelli et al, 2013).*

P2 R50 - To which grid do authors refer at this point of the article?

*The grid for Dust and Secondary Sulfate aerosol is 1° x 1°.*

P3 R70 – Is the "residence time" a property of each cell particle? Does it represent the time elapsed by each air parcel over each grid cell?

*Yes, they represent the time spent by the air parcels over each grid cell. Each 3 hours 40000 air parcels are released from each measurement station and are followed backward in time for 20 days. The residence time for each of these air parcels over each grid cell is calculated. Then the average is taken for all air parcels for each grid cell. This is the sensitivity for each 3 hours. We then sum these 3-hour sensitivities so as to correspond exactly to each filter sampling time.*

*We will add in the manuscript, in section 2.2 Flexible Particle Dispersion Model (FLEXPART):*

*The residence time for each of these air parcels over each grid cell is calculated. Then the average is taken for all air parcels for each grid cell. This is the sensitivity for each 3 hours. We then sum these 3-hour sensitivities so as to correspond exactly to each filter sampling time.*

P3 R75 – How is the pollution concentration associated to each air parcel?

*The sum for all grid cells of the {residence time in each grid cell (seconds) multiplied by the emissions in the grid cell (kg/(m$^2$\*seconds)) divided by 500 m (this is the height up to which we sum the residence time of each air parcel)}= concentration measured at station (kg/m$^3$)*

*The pollution concentration in each air parcel is linearly related to residence times and emissions*

*This is the forward problem.*

*We will add this in the manuscript in section 3.1 (Secondary Sulfate aerosol) as equation 5, in order to calculate model concentrations based on emission fluxes derived by Tikhonov regularization.*

P3-Figure 1  - What are the urban and suburban stations?

*Ankara and Belgrade stations are reported as suburban background by Almeida et al, 2020. Ankara station data are not used in the study, while Belgrade station data are used in both Secondary Sulfate and Dust aerosol analysis.*

*We will add in the manuscript in section 2.1 PM sampling stations and filter analysis:*

*Ankara and Belgrade stations are reported as suburban background by Almeida et al, 2020, while all other stations are reported as urban background.*

P5 R105-7  - What do "m" and "n" refer in this application?

*In lines 129-130 we write:*

*$m_{ij}$ is the sum of residence times (sensitivity) in a cell for concentrations higher than the 90$^{th}$ percentile and  $n_{ij}$ is the sum of residence times for all measurements.*

P6 R128 – are i and j the grid cell indexes?

*Yes, indexes corresponding to latitude and longitude of a grid cell.*

*We will add in the manuscript in section 2.5, Potential source contribution function (PSCF):*

*Indexes i,j  correspond to latitude and longitude of each grid cell.*

P6 R129 – When the authors state: "for concentrations higher than the 90$^{th}$ percentile" do they refer to air parcels generated in days with observed concentrations at the receptor higher than the 90$^{th}$ percentile?

*We will add in the manuscript, in section 2.5 Potential source contribution function (PSCF):*

*where $m_{ij}$ is the sum of residence times (sensitivity) in a cell with observed concentrations at the receptor higher than the 90$^{th}$ percentile*

P6 R130-133 – The role and the use of the weight matrix is not clear

*Polissar et al, 2001 report:*

*Since the PSCF is computed as a ratio of the counts of selected events ($m_{ij}$) to the counts of all events ($n_{ij}$), it is likely that relatively small $m_{ij}$ ($<n_{ij}$), which are often related to sparse trajectory coverage of the more distant grid cells, may result in $PSCF_{ij}$ with high uncertainty in the apparent high value. For large values of n, there is more statistical stability in the calculated value. Thus, to reduce the effect of small values of $n_{ij}$, an arbitrary weight function $W(n_{ij})$ is multiplied into the PSCF value to better reflect the uncertainty in the values for these cells.*

*We will add in the manuscript in section 2.5 Potential source contribution function (PSCF):*

*Grid cells with very small residence time may result in PSCF with high uncertainty in the apparent high value. For large values of $n_{ij}$, there is more statistical stability in the calculated value. Thus, to reduce the effect of small values of $n_{ij}$, an empirically determined weight matrix is multiplied into the PSCF value to better reflect the uncertainty in the values for these cells (Polissar et al, 2001).*

P6 R143 – The term "source" referred to Secondary sulfate detected with PMF analysis is misleading because it refers to a concentration not to a source. Maybe it could be referred as "source contribution to secondary sulfate concentration"

*We will add in the manuscript at the end of the Introduction section:*

*From now on, we refer to "source apportioned concentration by PMF" as "concentration" and to "geographic grid cell source areas emission fluxes" as "emission fluxes".*

P6 R151-152 – The term "source" is used again to indicate a contribution to concentration, I would suggest to thoroughly check the terminology

*Same as previous point.*

P7 R160 – The comparison between Figure 1 (a) and (d) should be expressed with harmonized units

*We added OMI-HTAP (EMISSIONS 2015 for SO$_2$) as suggested by referee #1. We will display in the manuscript the OMI-HTAP emission map in $kg*m^2*s^{-1}$ in Figure 3.*

P9 R176 – How is computed the modeled concentration? applying the FLEXPART model forward in time?

*Model concentration(kg/m$^3$) = residence time in each grid cell (seconds) * emission map deducted by Tikhonov regularization (kg/m$^2$*s$^{-1}$) divided by 500 m (height we consider relevant for emissions to take place)*

*We will add in the manuscript how this is produced (equation 5).*

*We state in lines 78-79:*

*Residence times in each grid cell, for a height range from 0 to 500 m above ground level (agl), are used for this study. The height was chosen so as to include sources within the boundary layer for all geographic grid cells.*

P10 R190 – why do authors state that dust is mainly of local origin? PSCF results in figure 5 seem to indicate a relevant role of long range transport (i.e. from Sahara)

*We state in line 191: The PSCF results for the rest of the stations indicated that their Dust aerosol was mainly of local origin (Dust re-suspension). We refer to stations (Ankara, Dushanbe, Vilnius, Krakow, Kurchatov, Banja-Luka, Chisinau, Niksic, Skopje, Sofia) whose PSCF analysis was not presented in the manuscript and does not indicate origin from a known Dust aerosol source.*

*We will add the names of the cities to the manuscript in section 3.2 Dust aerosol (Ankara, Dushanbe, Vilnius, Krakow, Kurchatov, Banja-Luka, Chisinau, Niksic, Skopje, Sofia). We will add the PSCF analysis for these cities as appendix A.4*

P10 R198-201 This statement is rather surprising considering that dust is non-reactive, therefore it is difficult to accept that "negative emissions" could be considered an acceptable solution

*Negative emissions could represent deposition velocity that is underestimated - especially in the beginning - by the FLEXPART deposition scheme. Also, this is due to inaccuracies in model and data*

*We will add in section 3.2 (Dust aerosol):*

*Stohl et al. (2009), referring to halocarbons, state that inaccuracies in model and data will in general cause their method to find solutions containing unrealistic negative emissions that are larger than expected. In the linear framework this cannot be prevented directly as positive definiteness is a nonlinear constraint. They also suggest an iteration method so as the sum of all negative emissions is less than 3‰ of the sum of the positive emissions. In our case with the Dust aerosol we allow small negative emission values (-2.5 * 10$^{-12}$ kg*m$^{-2}$*s$^{-1}$) representing higher deposition velocities than calculated by the FLEXPART deposition scheme.*

*References:*

Almeida, S., Manousakas, M., Diapouli, E., Kertesz, Z., Samek, L., Hristova, E., Šega, K., Alvarez, R. P., Belis, C., and Eleftheriadis, K.: Ambient particulate matter source apportionment using receptor modelling in European and Central Asia urban areas, Environmental Pollution, 266, 115 199, https://doi.org/10.1016/j.envpol.2020.115199, 2020.

Donatelli, M. and Reichel, L.: Square smoothing regularization matrices with accurate boundary conditions, Journal of Computational and Applied Mathematics, 272, 334–349, https://doi.org/10.1016/j.cam.2013.08.015, 2014.

Polissar, A. V., Hopke, P. K., and Harris, J.M.: Source Regions for Atmospheric Aerosol Measured at Barrow, Alaska, Environmental Science & Technology, 35, 4214–4226, https://doi.org/10.1021/es0107529, 2001.

Rodhe, H.: Budgets and turn-over times of atmospheric sulfur compounds, Atmospheric Environment (1967), 12, 671–680, 370 https://doi.org/10.1016/0004-6981(78)90247-0, 1978.

Stohl, A., Seibert, P., Arduini, J., Eckhardt, S., Fraser, P., Greally, B. R., Lunder, C., Maione, M., Mühle, J., O'Doherty, S., Prinn, R. G., Reimann, S., Saito, T., Schmidbauer, N., Simmonds, P. G., Vollmer, M. K.,Weiss, R. F., and Yokouchi, Y.: An analytical inversion method for determining regional and global emissions of greenhouse gases: Sensitivity studies and application to halocarbons, Atmospheric Chemistry and Physics, 9, 1597–1620, https://doi.org/10.5194/acp-9-1597-2009, 2009.

---

## Editor Decision (ED1)

General comments

1) I am not in favor of capitalizing Secondary Sulfate and Dust aerosol. These are not fixed terms or names and thus should be written in lower case letters.

2) Figure captions: (i) Please make one figure caption per figure, even if the figures include multiple panels (a, b, c etc), instead of having a caption under each panel.
(ii) if there is only one panel (e.g. Figure 8), there is no need to have a description including (a).
(iii) please improve all figure captions such that they describe what is shown and can be understood without having to read the full manuscript text (e.g. Figure 8 is much too brief)
(iv) Make sure that all figures are accessible for readers with vision deficiencies; check them with COBLIS
https://www.color-blindness.com/coblis-color-blindness-simulator/
Rainbow scale color scheme should be avoided to have unambiguous contrasts. Consider using different symbols if possible, e.g. in Figure 9.

3) Please provide a data availability statement according to the journal requirements
https://www.atmospheric-chemistry-and-physics.net/policies/data_policy.html#data_availability

Specific comments

l. 27: The sentence 'This work is the follow up of the article by Almeida et al.(2020).' seems out of place here and can be deleted. It may be better placed in the next sentence, e.g. by replacing 'in the aforementioned publication' by 'as identified by Almeida et al (2020)'.

l. 71: Please add a reference to EN12341

l. 73: Please add a reference to EPA PMF5.0

l. 74-76: Some text seems redundant here stating that only 50 samples were available for the six cities.

l. 92: what does the '(sensitivity)' here refer to? Can the word be simply deleted?

l. 101: do you mean 'included in the model'?

l. 125: Sometimes, you use 'we', sometimes 'the authors'... Using 'we' is more common and should be used for consistency (or replaced by passive voice).

l. 172: Please a reference for the report.

l. 176: 'Smaller differences...' sounds odd after the preceding text of '0 differences' (a difference < 0?). Please clarify.

l. 213: do you mean 'multiplied with'?

l. 224: replace 'that' by 'for which'

---

## Author Response (AR2)

**Suggestions for revision or reasons for rejection (visible to the public if the article is accepted and published)**

Authors introduced relevant updates and improvement with respect to both methodological description and presentation of the obtained results.

However the comparison of modelled concentrations (A*x) against observations point out a clear and systematic underestimation.

Authors state that such underestimation indicates that "most of observed concentrations are from local sources, hence they cannot be reproduced by the A*x model.

In my opinion this explanation is rather weak and questionable.

I would ask the authors to review the conclusions and partially the discussion section in order to:

- Better explain and justify the underestimation that affect modelled results

- Discuss which could be, anyhow, the usefulness of such model, although it shows a clear underestimation. This latter answer could take advantage of the arguments of the first answer.

*The authors are really thankful to the reviewer for the excellent comments and recommendations.*

*An error was found in the estimation of the OMI-HTAP emission map 1°\*1° average, which is now corrected.*

*Our answers follow in italics:*

1. Better explain and justify the underestimation that affect modelled results

*We will update the Tikhonov regularization equation in the manuscript, including the a priori information on emission fluxes from each grid cell ($x_0$). The equation follows:*

$$min\{||Ax-b||^2 + \lambda^2||L(x-x_0)||^2\}$$

*When no a priori information is available, we make the assumption in the Tikhonov regularization equation that $x_0$ is 0. In other words, our a priori information is that the emission fluxes from all grid cells are 0. We seek in our case a smooth solution, requesting that emission fluxes of neighboring cells have 0 differences. Smaller differences are achieved when absolute emission fluxes values are small (closer to 0). This imposes solutions with as small as possible emission fluxes, leading to the underestimation of measured values. The underestimation is relevant to how much we have to depart from the perfect fit (the fidelity term) of the measured data in the regularization procedure, which is achieved when the regularization parameter $\lambda$ is equal to 0. As we mention in section 2.1 (PM sampling stations and filter analysis), due to the high number of cities involved in the study, it was not possible to fully harmonize the methods used. In section 2.3 (Tikhonov regularization) we mention that due to positive and negative artifacts in the filter sampling, uncertainties in the filter analysis and PMF, an overall uncertainty approximating 30% is expected. In order to regularize such an uncertainty level, a large regularization parameter $\lambda$ is required. Thus, the regularization term is very important, leading to an underestimation of the resulting emission fluxes. We have to keep in mind that if $\lambda$ is close to 0, we perfectly reconstruct the measured concentrations, which include a*

*large error due to the reasons mentioned earlier. As the inverse process is not linear, such an approach would result in very large errors in the estimation of emission fluxes in each grid cell.*

*We will update in the manuscript in section 2.3 (Tikhonov regularization) the adjusted equation and replace lines 162-168 with the following:*

*"When no a priori information is available, the assumption in the Tikhonov regularization equation is that $x_0$ is 0. We seek in our case a smooth solution, requesting that emission fluxes of neighboring cells have 0 differences.  Smaller differences are achieved when absolute emission fluxes values are small (closer to 0). This imposes solutions with as small as possible emission fluxes, leading to the underestimation of measured values. The underestimation is relevant to how much we have to depart from the perfect fit (the fidelity term) of the measured data in the regularization procedure, which is achieved when the regularization parameter λ is equal to 0. As we mentioned in the previous paragraph, an overall uncertainty approximating 30% is expected. In order to regularize such an uncertainty level a large regularization parameter λ is required, thus leading to a significant underestimation of the model results. We have to keep in mind that if λ is close to 0, we perfectly reconstruct the measured concentrations, which include a large error due to the reasons mentioned earlier. As the inverse process is not linear, such an approach would result in very large errors in the estimation of emission fluxes in each grid cell."*

2. Discuss which could be, anyhow, the usefulness of such model, although it shows a clear underestimation. This latter answer could take advantage of the arguments of the first answer.

*The new method was developed in order to contribute to the air quality management process.*

*The usefulness of the proposed two-step method lies in the following points:*

1. *Improved identification of source areas for the long range transported aerosol in comparison to PSCF analysis.*
2. *Classification of relative importance of emission fluxes from geographic grid cells. This classification could be compared to existing emission inventories, resulting in possible improvements in the emissions calculation algorithms.*
3. *Estimation of the magnitude of emission fluxes from each grid cell. For Secondary Sulfate, around 60% of the measured concentrations magnitude could be reconstructed based on the deducted emission fluxes, while for Dust, approximately 45% could be reconstructed. This indicates that in this case, the new method significantly underestimates emission fluxes and measured concentrations. We have to keep in mind though that if data with much lower uncertainty are used, the underestimation would be significantly lower. Also, additional a priori information could lead to better performance of the method.*
4. *Since we identify the pollutant source area, its relative magnitude and acquire an estimate of the measured concentrations reconstruction, we can implement targeted mitigation measures. This approach can be used for any pollutant that can be simulated in FLEXPART or any similar model, without the need of an emission inventory.*
5. *Ideally, we would like to use the new method in combination to chemical transport models, so as to improve mitigation measures estimation. We should keep in mind that the emission fluxes deducted by the new method are averages over a period of 3 years. Emission fluxes have seasonal, monthly, weekday and daily variations in each grid cell that could not be*

*identified. Therefore, the emission fluxes result derived by the new method can only approximate roughly the concentrations measured at the cities participating in the study.*

*We will add in the manuscript as Appendix A5 the figures for Secondary Sulfate and Dust, where slopes between modeled and measured concentration without intercept are depicted. We will mention these figures in section 4 (Summary and Conclusions).*

*We will also add in section 4 (Summary and Conclusions, lines 393-409 in the revised manuscript) the following:*

*"The authors developed the new method in order to contribute to the air quality management process.*

*With this new method, an improved identification of source areas for the long range transported aerosol in comparison to simple PSCF analysis is achieved. Also, the relative importance of emission fluxes from each geographic grid cell is classified. This classification could be compared to existing emission inventories, resulting in possible improvements in the emissions calculation algorithms.*

*The new method also provides an estimate of the magnitude of emission fluxes from each grid cell. For Secondary Sulfate, around 60% of the measured concentrations magnitude could be reconstructed based on the deducted emission fluxes, while for Dust, approximately 45% could be reconstructed. This indicates that in this case, the new method significantly underestimates emission fluxes and measured concentrations. We have to keep in mind though that if data with much lower uncertainty are used, the underestimation would be significantly lower. Also, additional a priori information could lead to better performance of the method.*

*Since we identify the pollutant source area, its relative magnitude and an estimate of the measured concentrations reconstruction, we can implement targeted mitigation measures. This approach can be used for any pollutant that can be simulated in FLEXPART or any similar model, without the need of an emission inventory.*

*Ideally, we would like to implement the new method in combination to chemical transport models, so as to improve mitigation measures estimation.*

*We should keep in mind that the emission fluxes deducted by the new method are averages over a period of 3 years. Emission fluxes have seasonal, monthly, weekday and daily variations in each grid cell that could not be identified. Therefore, the emission fluxes result derived by the new method can only approximate roughly the concentrations measured at the cities participating in the study."*

---

## Author Response (AR3)

*The authors are grateful to the editor for comments and suggestions.*

*Our answers follow in italics:*

General comments
1) I am not in favor of capitalizing Secondary Sulfate and Dust aerosol. These are not fixed terms or names and thus should be written in lower case letters.

*They are changed to lowercase letters in the new version of the manuscript.*

2) Figure captions: (i) Please make one figure caption per figure, even if the figures include multiple panels (a, b, c etc), instead of having a caption under each panel.

*There is now one caption under each panel.*

(ii) if there is only one panel (e.g. Figure 8), there is no need to have a description including (a).

*This correction is applied in the manuscript.*

(iii) please improve all figure captions such that they describe what is shown and can be understood without having to read the full manuscript text (e.g. Figure 8 is much too brief)

*The figure captions are improved in the manuscript.*

(iv) Make sure that all figures are accessible for readers with vision deficiencies; check them with COBLIS https://www.color-blindness.com/coblis-color-blindness-simulator/
Rainbow scale color scheme should be avoided to have unambiguous contrasts. Consider using different symbols if possible, e.g. in Figure 9.

*The color palettes used are changed so as to be accessible by readers with vision deficiencies. I used different symbols when possible.*

3) Please provide a data availability statement according to the journal requirements

https://www.atmospheric-chemistry-and-physics.net/policies/data_policy.html#data_availability

*I have uploaded the necessary files and I acquired the following DOI:*
*10.5281/zenodo.7912792*

Specific comments
l. 27: The sentence 'This work is the follow up of the article by Almeida et al.(2020).' seems out of place here and can be deleted. It may be better placed in the next sentence, e.g. by replacing 'in the aforementioned publication' by 'as identified by Almeida et al (2020)'.

*This is now changed in the manuscript as suggested.*

l. 71: Please add a reference to EN12341

*A reference was added:*

*EN12341: Determination of the PM10 fraction of suspended particulate matter - Reference method and field test procedure to demonstrate reference equivalence of measurement methods., Tech. rep., CEN, 1998.*

l. 73: Please add a reference to EPA PMF5.0

*A reference was added:*
*Brown, S. G., Eberly, S., Paatero, P., and Norris, G. A.: Methods for estimating uncertainty in PMF solutions: Examples with ambient air and water quality data and guidance on reporting PMF results, Science of The Total Environment, 518–519, 626–635, https://doi.org/10.1016/j.scitotenv.2015.01.022, 2015.*

l. 74-76: Some text seems redundant here stating that only 50 samples were available for the six cities.

*A redundant sentence was removed.*

l. 92: what does the '(sensitivity)' here refer to? Can the word be simply deleted?

*The word '(sensitivity)' was removed.*

l. 101: do you mean 'included in the model'?

*It is changed in the manuscript from 'included in the results' to ' included in the model'.*

l. 125: Sometimes, you use 'we', sometimes 'the authors'... Using 'we' is more common and should be used for consistency (or replaced by passive voice).

*We have replaced 'the authors' in the manuscript with 'we', or passive voice, whenever applicable.*

l. 172: Please a reference for the report.

*We have added references for the AIRUSE project.*

l. 176: 'Smaller differences...' sounds odd after the preceding text of '0 differences' (a difference < 0?). Please clarify.

*We have rephrased lines 175-177:*
*'We seek in our case a smooth solution, requesting that emission fluxes of neighboring cells have differences close to 0, while at the same time the measured concentrations are reconstructed by the solution.  Solutions with small emission fluxes absolute values have smaller differences in neighboring cells than solutions with large emission fluxes absolute values. This imposes solutions with emission fluxes as small as possible, leading to the underestimation of measured values.'*

l. 213: do you mean 'multiplied with'?

*We have rephrased in the manuscript "multiplied into" to "multiplied with".*

*"Thus, to reduce the effect of small values of $n_{i;j}$ , an empirically determined weight matrix is multiplied with the PSCF value to better reflect the uncertainty in the values for these cells (Polissar et al., 2001)."*

l. 224: replace 'that' by 'for which

*We have replaced in the manuscript "that" to "for which".*

*"In appendix A1 we display the PSCF results for the rest of the cities for which a secondary sulfate concentration was identified."*